# Point-Focused Attention Meets Context-Scan State Space: Robust Biological Visual Perception for Point Cloud Representation

**Kanglin Qu**[1,2], **Pan Gao**[1,2*], **Qun Dai**[1,2*] **and Yuanhao Sun**[3]

[1]College of Artificial Intelligence, Nanjing University of Aeronautics and Astronautics
[2]Key Laboratory of Brain-Machine Intelligence Technology, Ministry of Education
[3]School of Mathematical Sciences, Beijing University of Posts and Telecommunications
`klinqu@163.com,{Pan.Gao,daiqun}@nuaa.edu.cn,sunyh@bupt.edu.cn`

## Abstract

Synergistically capturing intricate local structures and global contextual dependencies has become a critical challenge in point cloud representation learning. To address this, we introduce PointLearner, a point cloud representation learning network that closely aligns with biological vision which employs an active, foveation-inspired processing strategy, thus enabling local geometric modeling and long-range dependency interactions simultaneously. Specifically, we first design a point-focused attention, which simulates foveal vision at the visual focus through a competitive normalized attention mechanism between local neighbors and spatially downsampled features. The spatially downsampled features are extracted by a pooling method based on learnable inducing points, which can flexibly adapt to the non-uniform distribution of point clouds as the number of inducing points is controlled and they interact directly with point clouds. Second, we propose a context-scan state space that mimics eye's saccade inference, which infers the overall semantic structure and spatial content in the scene through a scan path guided by the Hilbert curve for the bidirectional S6. With this focus-then-context biomimetic design, PointLearner demonstrates remarkable robustness and achieves state-of-the-art performance across multiple point cloud tasks. The code is available at https://github.com/Point-Cloud-Learning/PointLearner.

## 1 Introduction

As a fundamental data form in 3D vision, point clouds have been widely applied in numerous tasks such as autonomous driving, robot navigation, and augmented reality due to their ability to precisely represent the geometric structure and spatial details of objects (Yan et al., 2024a; Zhou et al., 2024a; Yan et al., 2024b; An et al., 2025; Resani & Nasihatkon, 2025; Zhou et al., 2025b; He et al., 2024; Xu et al., 2024; Zhou et al., 2025a; Zhang et al., 2023; 2024a; Liang et al., 2025b; 2026). Currently, local attention networks (Zhao et al., 2021; Wu et al., 2024a) represent the mainstream paradigm for point cloud representation learning. By ingeniously computing attention within local neighbors/windows, they successfully reduce computational complexity to a linear relationship with the number of input points. However, such networks inevitably narrow the perceptual field, sacrificing the global perception capability of the attention mechanism, thereby hindering the effective modeling of long-range dependencies between objects in a scene. Recently, inspired by the exceptional long-range modeling capability with linear complexity of the selective state space model (S6) in Mamba (Gu & Dao, 2023), several studies (Zhang et al., 2024b; Han et al., 2024; Köprücü et al., 2024; Schöne et al., 2024; Wang et al., 2024) have attempted to introduce it into point cloud representation learning to overcome the trade-off between long-range interactions and computational resources. However, the bidirectional S6 still relies on compressing all contextual information into the history-hidden state for global connectivity, resulting in insufficient locality learning. Based on the above analysis, how to synergistically capture local fine-grained structures and global contextual dependencies has become a critical challenge in point cloud representation learning.

---

*Corresponding author.

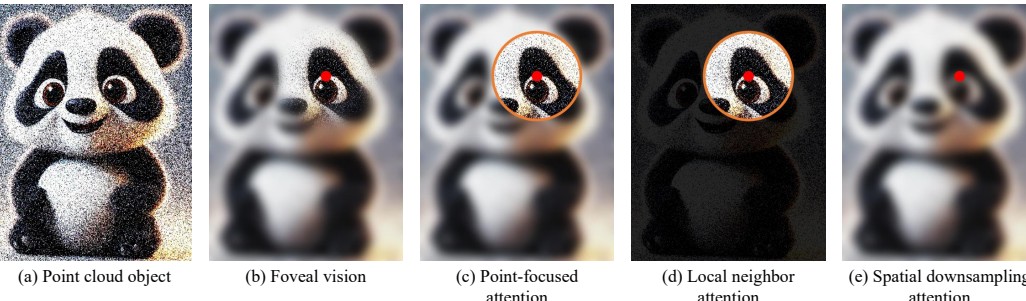

(a) Point cloud object     (b) Foveal vision     (c) Point-focused attention     (d) Local neighbor attention     (e) Spatial downsampling attention

Figure 1: A comparison of different information perception ways, where the red dot indicates the visual focus and the orange circle denotes the local awareness range. Our point-focused attention aligns more closely with natural foveal vision than the perceptual modes in (d) and (e).

In the biological visual system, foveal vision is responsible for perceiving the most significant information within the visual field, exhibiting pronounced spatial non-uniformity: as shown in Fig. 1(b), the region near the visual focus possesses extremely high acuity, enabling fine discrimination of detailed features; while visual acuity decreases with increasing eccentricity, resulting in coarser processing of peripheral features. This pattern not only optimizes the utilization of finite neural resources but also aligns with the intrinsic structure of information distribution in natural scenes (Wandell, 1995). What is more, biological vision is not a static process rather than a dynamic one that acquires information on a series of serialized visual foci via eye's continuous saccade movements, thereby inferring the overall semantic structure and spatial content within a scene (Stewart et al., 2020). The above operational mechanism endows the biological visual system with powerful perceptual capability to synergistically model local geometries and long-range dependencies.

Inspired by this, we explore PointLearner, a bionic-designed network that closely aligns with biological vision, enabling local geometric modeling and global context awareness simultaneously. In our network, the proposed point-focused attention emulate foveal vision perception at the visual focus, and context-scan state space is used to implement inference process during eye saccade:

- The point-focused attention adopts a dual-branch design, where one branch performs fine-grained attention modeling for each point's local neighbors, while the other branch establishes coarse-grained attention relationships between each point and spatially downsampled features. By computing attention weights from both branches within a single softmax calculation, the point-focused attention couples coarse- and fine-grained features in a competitive normalized manner with linear complexity. This makes each point adaptively and efficiently fuse local structures with global semantics, aligning with the intrinsic information distribution in natural scenes — like foveal vision perception. Notably, we develop an induced point pooling method, which is able to flexibly adapt to the non-uniform distribution of point clouds, through trainable vectors (dubbed inducing points) directly performing attention interactions with data points, as well as a controllable number of inducing points, thereby effectively downsampling spatial features.

- The context-scan state space utilizes the Hilbert curve for serialization of the point-focused attention feature, and further employs the the bidirectional S6 for geometric inference. As demonstrated in the eye tracking machine vision experiments (Newport et al., 2023), the Hilbert curve is more aligned with eye's saccade inference, having the property of locality preservation and the peculiarity of its self-similar rotating replication. By employing a biological inspired focus-then-context pipeline, the overall semantic structure and fine-grained spatial content can be efficiently integrated for point cloud representation learning.

To validate the effectiveness of PointLearner, we conduct extensive experiments on multiple standard point cloud datasets, including ModelNet40 (Wu et al., 2015), ScanObjectNN (Uy et al., 2019), ShapeNet (Yi et al., 2016), and S3DIS (Armeni et al., 2016). Experimental results demonstrate that our network, by emulating the operational mechanism of biological vision, effectively captures detail-rich local structures and coherent global context. It exhibits remarkable robustness and point cloud representation capability, yielding state-of-the-art results across multiple point cloud tasks.

In summary, the contributions of this paper stem from the following aspects:

(1) We propose PointLearner for point cloud learning, a bottom-to-up framework that aligns with biological vision, which achieves local refinement modeling and long-range dependency interactions by natural visual perception. This bio-inspired network attains state-of-the-art results on various point cloud tasks and demonstrates significant robustness.

(2) We design a point-focused attention simulating foveal vision perception at the visual focus. By computing attention weights for a point's local neighbors and spatially downsampled features within a single softmax calculation, it enables each point to adaptively and efficiently select the most effective receptive field information from both local structures and global semantics via a competitive normalized attention mechanism with linear complexity.

(3) We introduce a context-scan state space mimicking eye's saccade inference. By the Hilbert curve with excellent locality-preserving property, it guides the bidirectional S6 to accurately infer the entire scene along a scanning path that maintains high-fidelity spatial proximity between points.

## 2 RELATED WORK

### 2.1 ATTENTION-BASED NETWORKS

The attention mechanism (Vaswani et al., 2017; Shi, 2024; Su et al., 2024) has been widespread in point cloud representation learning, due to its ability to enable dynamic interactions between elements and global modeling. Several studies have enhanced the performance of the attention mechanism in point cloud tasks by refining attention modules (Guo et al., 2021; Mazur & Lempitsky, 2021; Yan et al., 2020) or designing pre-training strategies (Chen et al., 2023; Yu et al., 2022; Pang et al., 2022; Liu et al., 2022; Qi et al., 2023). Although these global attention networks have achieved impressive results, they perform attention computations directly on the entire point cloud, regarding each point as a token. This incurs prohibitive computational overheads due to the quadratic complexity of the attention mechanism and the large number of points in point clouds. Thus, some works ingeniously design local attention networks, which can be categorized into local neighbor-based and window-based methods. Local neighbor-based methods (Zhao et al., 2021; Wu et al., 2022; Nie et al., 2022; Xiang et al., 2023; Zhang et al., 2022; Liu et al., 2024b) apply the attention mechanism to point neighborhoods constructed for each point using neighbor search techniques such as the K-Means, ball query, and K-nearest neighbors (KNN), while window-based methods (Lai et al., 2022; Park et al., 2022; He et al., 2022; Fan et al., 2022; Sun et al., 2022; Liu et al., 2023b; Wu et al., 2024a) partition the 3D space into non-overlapping windows through voxelization or space-filling curves, transforming attention computations on all points to these spatial windows. Although local attention networks exhibit linear complexity, setting on the number of neighbors or window size constrains the receptive field of the attention mechanism, hindering the modeling of long-range dependencies. Departing from local neighbors, this paper incorporates a spatial downsampling branch and context-scan state space to extract global dependencies with the biological visual mechanism, overcoming the trade-off between global modeling and computational resources.

### 2.2 SSM-BASED NETWORKS

With excellent long-range modeling capability with linear complexity, the state space model (SSM) (Gupta et al., 2022; Gu et al., 2022; Smith et al., 2023; Mehta et al., 2023; Gu et al., 2020; 2021; Gu & Dao, 2023) has gained significant prominence in the field of natural language processing (NLP). Among these, Mamba (Gu & Dao, 2023) stands out as the most influential work. Its core component called S6 introduces an input-driven selective mechanism, enabling flexible selection of relevant information and achieving breakthrough performance in long sequence modeling. Furthermore, S6 incorporates a hardware-aware algorithm inspired by FlashAttention (Dao et al., 2022), significantly improving both training and inference efficiencies. These outstanding properties have motivated the extension of S6 from NLP to computer vision, including image recognition (Liu et al., 2025; Shaker et al., 2025; Fu et al., 2025; He et al., 2025), video understanding (Chen et al., 2025; Wang et al., 2023; Li et al., 2024a), and medical image segmentation (Xing et al., 2024; Ma et al., 2024; Ruan & Xiang, 2024). Recently, S6 has also been applied to point cloud representation learning. To adapt to the causal nature and unidirectional modeling of S6, existing methods design serializations strategies based on space-filling curves (Liang et al., 2024; Li et al., 2025; Liu et al., 2024a) or axis ordering (Zhang et al., 2024b; Han et al., 2024; Köprücü et al., 2024; Schöne et al., 2024)

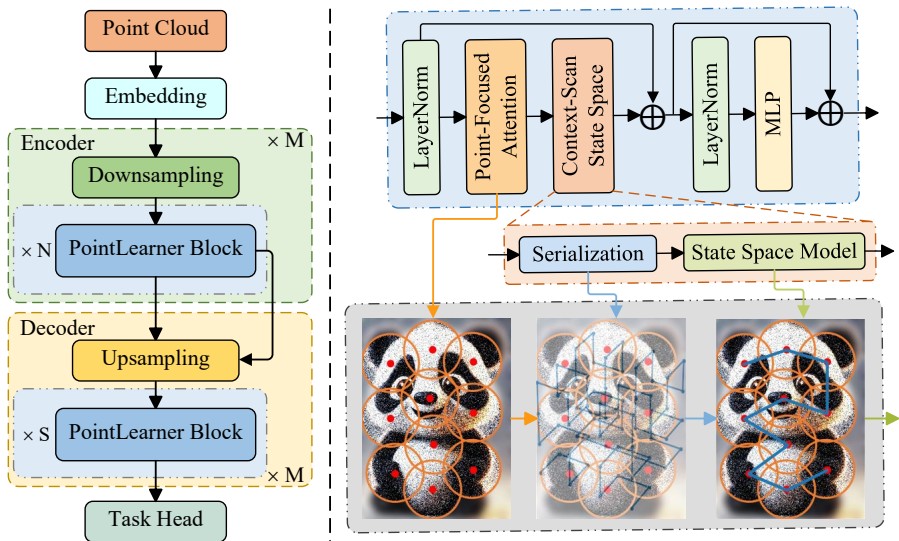

Figure 2: **Left**: Pipeline of PointLearner. **Right**: Architecture of PointLearner block, where the line between the red dots represent the saccade path guided by the serialization, which is used for geometric inference by the state space model.

to establish inter-point structural dependencies for the bidirectional S6's geometric inference. However, the bidirectional S6 still relies on compressing all context information into the history-hidden state to achieve global connectivity, leading to insufficient locality learning. Our work captures local structure information by a point-focused attention that matches foveal vision at the visual focus, addressing the above shortcoming.

## 3 METHODOLOGY

### 3.1 OVERVIEW

As shown in Fig. 2, PointLearner follows the standard Point Transformer-style architecture (Zhao et al., 2021; Wu et al., 2024a). A point cloud is first fed into an embedding layer formed by an MLP to be projected into a high-dimensional space, followed by an encoder-decoder structure that performs residual-based hierarchical feature aggregation, where the downsampling and upsampling layers use the Farthest Point Sampling (FPS) and linear interpolation (Qi et al., 2017b), respectively, and finally an appropriate task head is invoked based on specific requirements. In this study, our network is validated on point cloud recognition and segmentation tasks, where the recognition head first applies average pooling to the encoder's output, then produces global category logits by an MLP; the segmentation head processes the decoder's output with an MLP to predict per-point category logits. As the core component of the encoder-decoder structure, PointLearner block adopts the Transformer architecture for flexible integration into the network. By incorporating the point-focused attention first and then context-scan state space, it endows the network with information perception capability akin to the biological vision system, achieving local geometric modeling and long-range dependency interactions simultaneously. We will elaborate on these key modules next.

### 3.2 POINT-FOCUSED ATTENTION

The point-focused attention adopts a dual-branch design to emulate foveal vision at the visual focus, where the local neighbor branch provides fine-grained perception near each query point, while the spatial downsampling branch simultaneously maintains each query point's coarse-grained awareness for global semantics, *i.e.*, high acuity at the focus and low acuity in the periphery.

**Fine-grained perception**. Given a point set $\boldsymbol{P} = \{\boldsymbol{p}_i = (x_i, y_i, z_i)\}_{i=1}^{N} \in \mathbb{R}^{N \times 3}$ with a corresponding point feature set $\boldsymbol{F} = \{\boldsymbol{f}_i\}_{i=1}^{N} \in \mathbb{R}^{N \times D}$, the **L**ocal **N**eighbor **B**ranch on a point $\boldsymbol{p}_i$ is

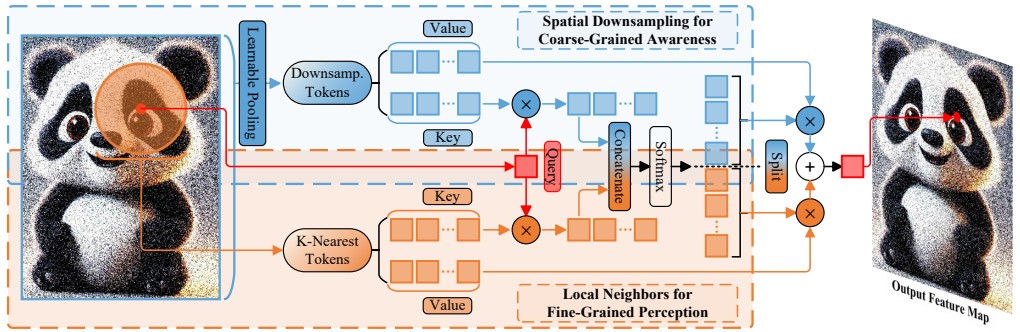

Figure 3: Diagram of point-focused attention with the competitive normalized attention mechanism.

computed as follows:

$$\left(\boldsymbol{Q}^l, \boldsymbol{K}^l, \boldsymbol{V}^l\right) = \left(\boldsymbol{W}_q^l, \boldsymbol{W}_k^l, \boldsymbol{W}_v^l\right)\boldsymbol{F}$$
$$\boldsymbol{\mathcal{A}}_i^l = softmax\left(\langle\boldsymbol{Q}_i^l, \boldsymbol{K}_{\boldsymbol{\mathcal{N}}_i}^l\rangle / \sqrt{D}\right) \tag{1}$$
$$\mathrm{LNB}\left(\boldsymbol{p}_i\right) = \boldsymbol{\mathcal{A}}_i^l \boldsymbol{V}_{\boldsymbol{\mathcal{N}}_i}^l$$

where $\boldsymbol{W} \in \mathbb{R}^{D \times D}$ represents a transformation matrix, and $\boldsymbol{\mathcal{N}}_i \in \mathbb{R}^K$ denotes the indices of the local neighbors of a point $\boldsymbol{p}_i$ in $\boldsymbol{P}$, determined by KNN.

**Coarse-grained awareness**. Given a point feature set $\boldsymbol{F} = \{\boldsymbol{f}_i\}_{i=1}^N \in \mathbb{R}^{N \times D}$ and its corresponding spatially downsampled feature set $\boldsymbol{S} = \{\boldsymbol{s}_i\}_{i=1}^M \in \mathbb{R}^{M \times D}$, the **S**patial **D**ownsampling **B**ranch on a point $\boldsymbol{p}_i$ is computed as follows:

$$\left(\boldsymbol{Q}^s, \boldsymbol{K}^s, \boldsymbol{V}^s\right) = \left(\boldsymbol{W}_q^s \boldsymbol{F}, \boldsymbol{W}_k^s \boldsymbol{S}, \boldsymbol{W}_v^s \boldsymbol{S}\right)$$
$$\boldsymbol{\mathcal{A}}_i^s = softmax\left(\boldsymbol{Q}_i^s, \boldsymbol{K}^s / \sqrt{D}\right) \tag{2}$$
$$\mathrm{SDB}\left(\boldsymbol{p}_i\right) = \boldsymbol{\mathcal{A}}_i^s \boldsymbol{V}^s$$

Notably, the non-uniformity of point clouds prevents them from achieving spatial downsampling through simple and effective average pooling like 2D images. Moreover, as a commonly used downsampling method for point clouds, FPS often requires setting a small sampling rate to attain sufficient coverage for extracting global information, which significantly increase computational cost in Eq. (2). Thus, we develop an induced point pooling inspired by the inducing point method in the sparse Gaussian (Snelson & Ghahramani, 2005). Specifically, $M$ $D$-dimensional vectors defined as $\boldsymbol{I} \in \mathbb{R}^{M \times D}$ are termed inducing points, and they are trainable parameters. The **I**nduced **P**oint **P**ooling on a point feature set $\boldsymbol{F} = \{\boldsymbol{f}_i\}_{i=1}^N \in \mathbb{R}^{N \times D}$ is computed as follows:

$$\left(\boldsymbol{K}^p, \boldsymbol{V}^p\right) = \left(\boldsymbol{W}_k^p, \boldsymbol{W}_v^p\right)\boldsymbol{F}$$
$$\boldsymbol{S} = \mathrm{IPP}\left(\boldsymbol{F}\right) = softmax\left(\boldsymbol{I}, \boldsymbol{K}^p / \sqrt{D}\right) \boldsymbol{V}^p \tag{3}$$

where $\boldsymbol{S} \in \mathbb{R}^{M \times D}$ represents the spatially downsampled features extracted by the induced point pooling on $\boldsymbol{F}$. By using a controllable number of trainable inducing points to directly learn how to induct point clouds adaptively through attention interactions, the induced point pooling flexibly adapts to the non-uniform distribution of point clouds to integrate global semantics, thereby effectively downsampling spatial features, as shown in Appendix C.2.

**Competitive normalized fusion**. A straightforward approach for the point-focused attention is to sum the attention outputs from both branches to achieve the multi-scale fusion PFA $\left(\boldsymbol{p}_i\right)$ of the local structures and global semantics at a point $\boldsymbol{p}_i$, thereby matching foveal vision perception, as follows:

$$\mathrm{PFA}\left(\boldsymbol{p}_i\right) = \mathrm{LNB}\left(\boldsymbol{p}_i\right) + \mathrm{SDB}\left(\boldsymbol{p}_i\right) = \boldsymbol{\mathcal{A}}_i^l \boldsymbol{V}_{\boldsymbol{\mathcal{N}}_i}^l + \boldsymbol{\mathcal{A}}_i^s \boldsymbol{V}^s \tag{4}$$

However, such *shallow* multi-scale feature fusion struggles to align with deep dynamic interactions between local fine-grained features and global coarse-grained semantics inherent in foveal vision

perception. To address this, as illustrated in Fig. 3, we update the point-focused attention to a competitive normalized attention variant by computing the attention weights of both branches within a single softmax calculation, thereby coupling fine- and coarse-grained features, as follows:

$$\boldsymbol{\mathcal{A}}_i = softmax\left(Concat\left(\boldsymbol{Q}_i^l, \boldsymbol{K}_{\mathcal{N}_i}^l, \boldsymbol{Q}_i^s, \boldsymbol{K}^s\right)/\sqrt{D}\right)$$

$$\boldsymbol{\mathcal{A}}_i^l, \boldsymbol{\mathcal{A}}_i^s = split\left(\boldsymbol{\mathcal{A}}_i, [K, M]\right) \tag{5}$$

$$\text{PFA}\left(\boldsymbol{p}_i\right) = \text{LNB}\left(\boldsymbol{p}_i\right) + \text{SDB}\left(\boldsymbol{p}_i\right) = \boldsymbol{\mathcal{A}}_i^l \boldsymbol{V}_{\mathcal{N}_i}^l + \boldsymbol{\mathcal{A}}_i^s \boldsymbol{V}^s$$

where $Concat$ denotes channel-level concatenation, $split$ serves to partition channels according to specified sizes, and $K$ and $M$ represent the number of local neighbors determined by KNN and the number of inducing points used for downsampling spatial features, respectively. Compared to the simple version in Eq. (4), as demonstrated in Tab. 8, Eq. (5) enhances deep dynamic interactions through the competitive mechanism between fine- and coarse-grained features, without introducing additional computational overheads. This effectively simulates the perceptual process of foveal vision adaptively selecting the most effective receptive field information, thereby better aligning with the intrinsic structure of information distribution in natural scenes.

**Complexity analysis**. Based on the above settings and considering the feature transformation, the computational complexity $\Omega\left(\text{PFA}\right)$ of the point-focused attention is as follows:

$$\Omega\left(\text{PFA}\right) = \Omega\left(\text{LNB}\right) + \Omega\left(\text{SDB}\right) + \Omega\left(\text{IPP}\right)$$

$$= 6ND^2 + 2MD^2 + 2NKD + 4NMD \tag{6}$$

Since $K$ and $M$ are typically small, it can be observed that $\Omega\left(\text{PFA}\right)$ scales linearly with the number of points, indicating that PFA aligns with the optimized utilization of finite neural resources in foveal visual. For the complexity of each module in PFA, please refer to Appendix D.

### 3.3 CONTEXT-SCAN STATE SPACE

The context-scan state space simulates eye's saccade inference through a dynamic saccade mechanism. It serializes a point cloud to provide a scanning path for the state space model, enabling the inference of the overall semantic structure and spatial content in a scene, *i.e.*, the serialization performs continuous scanning and the state space model is responsible for information integration, as illustrated in Fig. 2 (right).

**Serialization**. Space-filling curves are trajectories that cover high-dimensional regions by continuous parametric mapping. Their core function is to transform high-dimensional geometric structures into one-dimensional sequences while preserving local neighborhood relationships, meaning that spatially adjacent elements remain adjacent in the sequence. When applied to point clouds, the high-dimensional space refers to the 3D Euclidean space containing point coordinates. Common space-filling curves include the Hilbert curve and Z-Order curve. The former is highly valued for its superior locality-preserving property, while the latter is renowned for its high efficiency.

Inspired by the spatial proximity of space-filling curves, recent S6-based works (Liang et al., 2024; Li et al., 2025; Liu et al., 2024a) utilize them to serialize point clouds, establishing more reliable inter-point structural dependencies for geometric reasoning compared to axis-ordering based serialization (Zhang et al., 2024b; Han et al., 2024; Köprücü et al., 2024; Schöne et al., 2024). However, to provide richer spatial information, these serialization strategies often concatenate the results from multiple space-filling curves. This concatenation introduces a longer sequence, leading to redundancy and negatively impacting efficiency. More importantly, concatenating sequences with different spatial relationships can easily cause confusion. In addition, by serializing randomly located points using the Hilbert and Z-Order curves, respectively, it can be intuitively observed that the Hilbert curve outperforms the Z-Order curve in preserving spatial proximity, as shown in Appendix C.1. This finding is consistent with the prior research (Nordin & Telles, 2023) and also better aligns with the inherent pattern of eye movements during visual search, which typically involve continuous scanning along spatially adjacent regions. Therefore, the Hilbert curve is employed for serialization to establish reliable inter-point structural dependencies while guiding a high-fidelity spatially adjacent scanning path for accurate scene inference.

**State space model**. S6 is a forward recurrence process based on hidden states, where each position in input sequence can only access prior information and cannot obtain content from subsequent positions, as shown in Appendix A. This unidirectional modeling is unsuitable for visual data requiring

| Network | Operator | OA |
|---|---|---|
| [†]IDPT (Zha et al., 2023) | Attention | 93.4 |
| [†]Inter-MAE (Liu et al., 2023a) | Attention | 93.6 |
| [†]CrossNet (Wu et al., 2023b) | Attention | 93.4 |
| [†]ACT (Dong et al., 2023) | Attention | 93.7 |
| LFT-Net (Gao et al., 2023) | Attention | 93.2 |
| OctFormer (Wang, 2023) | Attention | 92.7 |
| [†]ReCon (Qi et al., 2023) | Attention | 92.5 |
| [†]DAPT (Zhou et al., 2024b) | Attention | 93.5 |
| [†]LCM (Zha et al., 2024) | Attention | 93.6 |
| [†]Point-PEFT (Tang et al., 2024) | Attention | 93.4 |
| PointStack (Wijaya et al., 2024) | Attention | 93.3 |
| GAD (Li et al., 2024b) | Attention | 93.8 |
| PointConT (Liu et al., 2024b) | Attention | 93.5 |
| [†]PointGST (Liang et al., 2025a) | Attention | 93.4 |
| [†]PointMamba (Liang et al., 2024) | SSM | 93.6 |
| OctMamba (Liu et al., 2024a) | SSM | 92.7 |
| PCM (Zhang et al., 2024b) | SSM | 93.4 |
| [†]Mamba3D (Han et al., 2024) | SSM | 93.4 |
| NIMBA (Köprücü et al., 2024) | SSM | 92.1 |
| STREAM (Schöne et al., 2024) | SSM | 92.7 |
| PoinTramba (Wang et al., 2024) | Hybrid | 92.7 |
| PointLearner | Hybrid | **94.2** |

Table 1: Experimental results on Model-Net40 dataset. [†]: Pre-training strategy.

| Network | Operator | Ins. mIoU |
|---|---|---|
| APES (Wu et al., 2023a) | Attention | 85.8 |
| [†]ACT (Dong et al., 2023) | Attention | 86.1 |
| [†]PointGPT (Chen et al., 2023) | Attention | 86.2 |
| [†]ReCon (Qi et al., 2023) | Attention | 86.4 |
| [†]Point2Vec (Zeid et al., 2023) | Attention | 86.3 |
| [†]IDPT (Zha et al., 2023) | Attention | 85.7 |
| GAD (Li et al., 2024b) | Attention | 86.3 |
| [†]MaskFeat3D (Yan et al., 2024b) | Attention | 86.3 |
| [†]MVNet (Yan et al., 2024a) | Attention | 86.1 |
| [†]LCM (Zha et al., 2024) | Attention | 86.3 |
| [†]DAPT (Zhou et al., 2024b) | Attention | 85.5 |
| [†]Point-PEFT (Tang et al., 2024) | Attention | 85.1 |
| [†]PointGST (Liang et al., 2025a) | Attention | 85.7 |
| [†]Mamba3D (Han et al., 2024) | SSM | 85.6 |
| [†]PointMamba (Liang et al., 2024) | SSM | 86.2 |
| PCM (Zhang et al., 2024b) | SSM | 84.3 |
| NIMBA (Köprücü et al., 2024) | SSM | 85.5 |
| PoinTramba (Wang et al., 2024) | Hybrid | 85.7 |
| PointLearner | Hybrid | **86.9** |

Table 2: Experimental results on ShapeNet dataset. [†]: Pre-training strategy.

global learning. To address this, most works (Li et al., 2025; Han et al., 2024; Köprücü et al., 2024; Wang et al., 2024) propose the bidirectional S6, by introducing the bidirectionality from Vision Mamba (Zhu et al., 2024) into S6, to achieve global modeling over point sequences. Specifically, two S6 modules are deployed in parallel: a forward S6 and a backward S6. The former performs a forward-scanning recurrent along the input sequence, while the latter processes it in reverse order. In this way, each point in the input sequence possesses a global receptive field, which aligns with how the eye perform back-and-forth scanning to infer information when recognizing an indistinct object. Hence, we adopt the bidirectional S6 for scene inference.

# 4 EXPERIMENTS

To validate PointLearner, we conduct experimental comparisons on multiple point cloud tasks. Additionally, we perform robustness checking, and explore the effectiveness and characteristics of these biomimetic designs through extensive ablation studies. For detailed descriptions on datasets and evaluation metrics, please refer to Appendix B.

## 4.1 EXPERIMENTAL COMPARISONS

**Object recognition**. Table 1 lists the quantitative results of our network and recent works on Model-Net40 dataset. It can be observed that the performance of the previous state-of-the-art attention networks has been saturated, confined to a narrow range of 93.2% to 93.8%. Our PointLearner breaks through this performance bottleneck, achieving a state-of-the-art 94.2% OA. This result demonstrates that integrating the advantages of both attention and SSM paradigms within a biological visual system framework is an effective avenue to advancing point cloud representation learning.

**Part segmentation**. Table 2 lists the quantitative results of our network and recent works on ShapeNet dataset. Overall, the performance of emerging SSM-based methods confirms their limitations in handling fine-grained local features. The proposed PointLearner achieves 86.9% Ins. mIoU, significantly outperforming existing state-of-the-art methods based on either attention or SSM. This strongly demonstrates that the hybrid paradigm of the attention and SSM, guided by the biological vision system, can effectively synergize the strengths of both operators, exhibiting powerful capabilities in local geometric modeling and long-range dependency interactions.

**Semantic segmentation**. Table 3 lists the quantitative results of our network and recent works on S3DIS dataset. In the more challenging task of point cloud semantic segmentation, the performance of the networks with different architectures exhibits significant disparities, highlighting the dual

| Network | Operator | mIoU |
|---|---|---|
| PointVector (Deng et al., 2023) | Attention | 72.3 |
| SPT (Robert et al., 2023) | Attention | 68.9 |
| SpoTr (Park et al., 2023) | Attention | 70.8 |
| [†]ACT (Dong et al., 2023) | Attention | 61.2 |
| [†]ReCon (Qi et al., 2023) | Attention | 60.8 |
| [†]IDPT (Zha et al., 2023) | Attention | 53.1 |
| [†]MM-3Dscene (Xu et al., 2023) | Attention | 71.9 |
| Retro-FPN (Xiang et al., 2023) | Attention | 73.0 |
| PTv3 (Wu et al., 2024a) | Attention | 73.4 |
| KPConvX-L (Thomas et al., 2024) | Attention | 73.5 |
| [†]DAPT (Zhou et al., 2024b) | Attention | 56.2 |
| [†]Point-PEFT (Tang et al., 2024) | Attention | 56.0 |
| [†]MVNet (Yan et al., 2024a) | Attention | 73.8 |
| GAD (Li et al., 2024b) | Attention | 62.9 |
| [†]Swin3D (Yang et al., 2025) | Attention | 72.5 |
| [†]PointGST (Liang et al., 2025a) | Attention | 58.6 |
| PCM (Zhang et al., 2024b) | SSM | 63.4 |
| HydraMamba (Qu et al., 2025) | SSM | 73.6 |
| Pamba (Li et al., 2025) | SSM | 73.5 |
| PointLearner | Hybrid | **74.3** |

Table 3: Experimental results on S3DIS dataset. [†]: Pre-training strategy.

| Network | Operator | OA |
|---|---|---|
| ADS (Hong et al., 2023) | Attention | 87.5 |
| [†]IDPT (Zha et al., 2023) | Attention | 84.9 |
| [†]ACT (Dong et al., 2023) | Attention | 88.2 |
| [†]Joint-MAE (Guo et al., 2023) | Attention | 86.1 |
| SpoTr (Park et al., 2023) | Attention | 88.6 |
| [†]LCM (Zha et al., 2024) | Attention | 87.8 |
| [†]Inter-MAE (Liu et al., 2023a) | Attention | 85.4 |
| [†]PointGPT (Chen et al., 2023) | Attention | 86.9 |
| [†]PointDif (Zheng et al., 2024) | Attention | 87.6 |
| [†]DAPT (Zhou et al., 2024b) | Attention | 85.1 |
| GAD (Li et al., 2024b) | Attention | 82.6 |
| [†]Point-PEFT (Tang et al., 2024) | Attention | 85.0 |
| [†]MaskFeat3D (Yan et al., 2024b) | Attention | 87.7 |
| [†]MVNet (Yan et al., 2024a) | Attention | 86.7 |
| [†]PointGST (Liang et al., 2025a) | Attention | 85.6 |
| [†]PointMamba (Liang et al., 2024) | SSM | 89.3 |
| PCM (Zhang et al., 2024b) | SSM | 88.1 |
| [†]Mamba3D (Han et al., 2024) | SSM | 88.2 |
| NIMBA (Köprücü et al., 2024) | SSM | 84.2 |
| STREAM (Schöne et al., 2024) | SSM | 85.3 |
| PoinTramba (Wang et al., 2024) | Hybrid | 88.9 |
| PointLearner | Hybrid | **89.8** |

Table 4: Experimental results on ScanObjectNN dataset. [†]: Pre-training strategy.

challenges of local geometric modeling and global contextual inference in this task. Our network attains state-of-the-art performance with 74.3% mIoU, which benefits from its bio-inspired visual perception design, maintaining the sensitivity of the attention mechanism to both local geometries and global semantics while incorporating the scene inference capability of SSM.

## 4.2 ROBUSTNESS CHECKING

Biological vision exhibits remarkable robustness in to low-quality scenes. To verify that our biomimetic network inherits this property, we conduct experiments from two perspectives: strong noise corruption and varying sampling densities.

**Robustness to strong noise**. ScanObjectNN is a challenging dataset collected from real-world scenes. We conduct experiments on its most difficult variant, PB_T50_RS, to validate PointLearner's robustness to strong noise. Tab. 4 lists the quantitative results of our network and recent works on ScanObjectNN dataset. The performance of the hybrid architecture PoinTramba (88.9% OA) has demonstrated the potential of hybrid methods in terms of robustness. Our PointLearner further elevates the performance to 89.8% OA, surpassing all the existing models. This result indicates that the synergistic mechanism of the attention and SSM adopted by PointLearner effectively inherits the robust inference capability of the biological vision system under low-quality perception conditions.

**Robustness to varying sampling densities**. Sensor data captured directly from the real world often suffers from severe irregular sampling issues. Consequently, during the testing phase, we randomly discard data points, as shown in Fig. 4 (left), to validate our network's robustness to non-uniformly sparse data. Fig. 4 (right) presents the quantitative results of our network alongside the top-performing attention method (GAD (Li et al., 2024b)) and SSM method (PCM (Zhang et al., 2024b)) from Tab. 1 on ModelNet40 dataset with varying numbers of sampling points. Intuitively, our network exhibits superior robustness to variations in sampling density compared to the other methods, with only a 2.2% performance drop when the number of test points decreases from 1024 to 256. This result benefits from the competitive normalized attention in the point-focused attention, which balances descriptiveness and robustness by appropriately weighting local structures and global perception, as well as the powerful global inference capability of the context-scan state space.

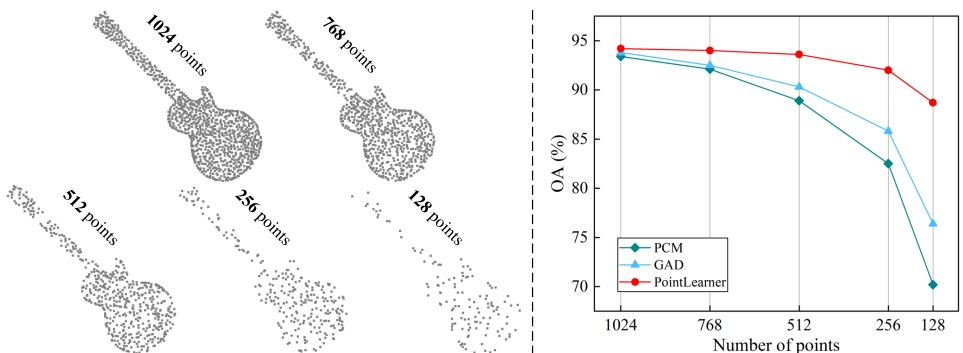

Figure 4: **Left**: Point clouds with different point densities. **Right**: Quantitative results of our network and other works on ModelNet40 dataset with different numbers of sampling points.

| Networks | Operator | Params | Latency | Memory | mIoU |
|---|---|---|---|---|---|
| HydraMamba (Qu et al., 2025) | SSM | 63.14M | 54ms | **5.9G** | 73.6 |
| PTv3 (Wu et al., 2024a) | Attention | 46.17M | **49ms** | 6.3G | 73.4 |
| Swin3D (Yang et al., 2025) | Attention | 71.15M | 365ms | 10.7G | 72.5 |
| PointLearner | Hybrid | 52.78M | 63ms | 6.5G | **74.3** |

Table 5: Params, latency, and memory footprint of our network and previous state-of-the-art methods in a single inference on S3DIS dataset.

## 4.3 EFFICIENCY ANALYSIS

To further analyze the computational overheads of PointLearner, we compared it with several previous state-of-the-art methods on S3DIS dataset, and the params, latency, and memory footprint in a single inference are selected as evaluation metrics. Specifically, to ensure a fair comparison, the latency and memory footprint in a single inference are taken as the average values obtained over the entire S3DIS test set with each scene on the same RTX 4090 GPU. Tab. 5 presents the params, latency, and memory footprint of our network and multiple previous state-of-the-art methods in a single inference. Swin3D is a *heavy* attention network that, unlike PTv3, does not incorporate efficiency improvements for attention computations. Furthermore, it can be observed that, with a similar number of parameters, PointLearner, as a hybrid network, achieves a superior trade-off between computational overheads and performance compared to the fully optimized PTv3 and the Hydra-Mamba that benefits from S6's excellent properties. We attribute this to the following two aspects: (1) PointLearner is able to achieve powerful local geometric modeling and long-range dependency interactions from the perspective of biological vision through hybrid operators, which simultaneously requires less layer stacking; (2) Although our basic block contains multiple components, they are essentially *lightweight* with linear complexity, while also benefiting from the hardware-optimized algorithms of FlashAttention (Dao et al., 2022).

## 4.4 ABLATION STUDY

We conduct ablation study on ModelNet40 dataset, to investigate the effectiveness of the designed bio-inspired visual processing modules. To ensure a fair comparison, all experiments are conducted on an RTX 4090 GPUs with identical configurations, and results are averaged over three runs.

**Local neighbor branch**. The point-focused attention employs the local neighbor branch to achieve the fine-grained perception, akin to foveal vision. To validate its necessity, we compare ablation results with and without the local neighbor branch in Tab. 6. The trade-off between accuracy and efficiency further indicates that the local neighbor branch enhances the network's ability to learn local geometric structures while maintaining low computational complexity. This closely aligns with the biological vision mechanism where refined local perception optimizes global inference.

**Spatial downsampling branch**. Through the spatial downsampling branch, the point-focused attention simultaneously maintains coarse-grained perception of global semantics for each query point. To validate its importance, we compare ablation results with and without this branch in Tab. 7.

| LNB | Params | FLOPs | Throughput | OA |
|-----|--------|-------|------------|-----|
| ✓ | 7.36M | 0.610G | 163FPS | **94.17** |
| × | 6.57M | 0.504G | 221FPS | 92.11 |

Table 6: Ablation results w/ or w/o the local neighbor branch.

| SDB | Params | FLOPs | Throughput | OA |
|-----|--------|-------|------------|-----|
| ✓ | 7.36M | 0.610G | 163FPS | **94.17** |
| × | 6.35M | 0.578G | 183FPS | 93.06 |

Table 7: Ablation results w/ or w/o the spatial downsampling branch.

| Fusion | Params | FLOPs | Throughput | OA |
|--------|--------|-------|------------|-----|
| Competitive | 7.36M | 0.610G | 163FPS | **94.17** |
| Additive | 7.36M | 0.610G | 166FPS | 93.43 |

Table 8: Ablation results with different multi-scale fusion.

| SSM | Params | FLOPs | Throughput | OA |
|-----|--------|-------|------------|-----|
| Bid. S6 | 7.36M | 0.610G | 163FPS | **94.17** |
| Uni. S6 | 6.06M | 0.553G | 181FPS | 93.08 |

Table 9: Ablation results with both state space models.

| PFA | CSSS | Params | FLOPs | Throughput | OA |
|-----|------|--------|-------|------------|-----|
| ✓ | × | 4.57M | 0.487G | 198FPS | 92.93 |
| × | ✓ | 5.37M | 0.463G | 231FPS | 91.94 |
| ✓ | ✓ | 7.36M | 0.610G | 163FPS | **94.17** |

Table 10: Ablation results with PFA and CSCC.

Quantitative analysis demonstrates that this branch effectively establishes coarse-grained associations between query points and global semantics at a low computational cost, simulating the integration mechanism of peripheral information in biological vision.

**Normalized fusion**. In the point-focused attention, by coupling attention weights within a single softmax calculation, the proposed competitive normalized fusion replaces simple additive multi-scale fusion to enhance deep dynamic interactions. To validate its advantage, Tab. 8 compares the ablation results of both multi-scale fusion strategies. Quantitative analysis demonstrates that the competitive fusion mechanism effectively simulates the dynamic interaction and adaptive selection between fine- and coarse-grained features in foveal vision. This achieves superior semantic fusion effect while maintaining nearly identical computational overheads.

**State space model**. The context-scan state space leverages the state space model to infer the entire scene along the scanning path provided by the serialization, based on inter-point structural dependencies. To validate that the adopted bidirectional S6 achieves stronger inference performance compared to the unidirectional S6, we compared the ablation results of both state space models. As shown in Tab. 9, although the bidirectional S6 introduces a slight increase in parameters and computational complexity, it constructs a global receptive field for each point through the forward and backward scanning. This effectively simulates the eye's back-and-forth saccade, thereby enhancing inference ability in complex scenes.

**PFA & CSSS**. The proposed network incorporates two key modules: Point-Focused Attention (PFA) and Contextual Scan State Space (CSSS). PFA fuses local neighbors and spatially downsampled features based on a competitive normalized attention mechanism, simulating foveal vision at the visual focus. On this basis, CSSS further infers the overall semantic structure and spatial content within a scene through point cloud serialization and state space model, mimicking eye's saccade inference. The combination of PFA and CSSS closely aligns with biological vision, and their complementarity is fully validated by the ablation results presented in Tab. 10.

## 5 CONCLUSION

In this paper, we introduce PointLearner, a point cloud representation learning network inspired by biological vision mechanisms. First, we propose a point-focused attention architecture that mimics the foveal vision of the human eye, enabling fine-grained perception around each query point while maintaining coarse-grained awareness of more distant regions. Building on this locally attentive representation, we further introduce a context-scan state space model to globally scan point clouds, drawing inspiration from the saccadic movements of the human eye during visual inference. This allows the model to integrate local details and global context, leading to a comprehensive understanding of the underlying geometric structure. Extensive experiments across multiple datasets and tasks demonstrate that PointLearner outperforms existing methods and exhibits strong robustness to noise and varying sampling densities.

ACKNOWLEDGMENTS

This work is supported in part by the National Natural Science Foundation of China under Grant (62476126, 62272227).

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

## A   STATE SPACE MODELS

SSMs are cyclic processes with latent states, which map a 1-D equation or sequence $x(t) \in \mathbb{R}^N$ to $y(t) \in \mathbb{R}^N$ by a latent state $h(t) \in \mathbb{R}^N$. The process is mathematically denoted as a linear ordinary differential equation as follows

$$y(t) = \boldsymbol{C}h(t), h'(t) = \boldsymbol{A}h(t) + \boldsymbol{B}x(t), \tag{7}$$

where the three parameters $\boldsymbol{A} \in \mathbb{R}^{N \times N}$, $\boldsymbol{B} \in \mathbb{R}^N$, and $\boldsymbol{C} \in \mathbb{R}^N$ represent the state matrix, input matrix, and output matrix, respectively. Since the above SSMs run on continuous inputs and are not applicable to discrete inputs such as images and text, they cannot be introduced into deep models. Thus, it is necessary to discretize them, and the zero-order hold is commonly used as a discretization method. The discretized formulas are as follows

$$h_t = \bar{\boldsymbol{A}}h_{t-1} + \bar{\boldsymbol{B}}x_t, \quad y_t = \boldsymbol{C}h_t, \tag{8}$$

where $\bar{\boldsymbol{A}}$ and $\bar{\boldsymbol{B}}$ are the results of discretizing the continuous parameters $\boldsymbol{A}$ and $\boldsymbol{B}$ by a time scale $\boldsymbol{\Delta}$, denoted as

$$\bar{\boldsymbol{A}} = e^{\boldsymbol{\Delta}\boldsymbol{A}}, \quad \bar{\boldsymbol{C}} = \boldsymbol{C}, \quad \bar{\boldsymbol{B}} = (\boldsymbol{\Delta}\boldsymbol{A})^{-1}(e^{\boldsymbol{\Delta}\boldsymbol{A}} - \boldsymbol{I})(\boldsymbol{\Delta}\boldsymbol{B}). \tag{9}$$

Since processing the input and latent state equally, previous approaches focusing on linear time-invariant SSMs (where $\bar{\boldsymbol{A}}$ and $\bar{\boldsymbol{B}}$ are invariant) may fail to capture critical information from context. Hence, Mamba proposes a novel SSM termed S6 by integrating an input-dependent selective mechanism into SSMs, where $\bar{\boldsymbol{A}}$ and $\bar{\boldsymbol{B}}$ are the functions of inputs, indicating Mamba is linear time-variant.

## B   DATASETS AND IMPLEMENTATION

### B.1   DATASETS

ModelNet40 dataset contains 12,311 CAD models across 40 categories, with 9,843 samples in the training set and 2,468 samples in the test set. Data preprocessing follows the method of Qi et al. (2017a), where 1,024 points and their normal vectors are uniformly sampled from each sample as input. As per most relevant studies in the literature, the overall accuracy (OA) is adopted as an evaluation metric.

ShapeNet dataset comprises 16,878 samples from 50 parts across 16 categories, with 14,005 samples in the training set and 2,873 in the test set. Data preprocessing is consistent with that applied to ModelNet40 dataset. As per most relevant studies in the literature, the instance mIoU (Ins. mIoU) is adopted as an evaluation metric.

S3DIS dataset comprises 3D point cloud data from 271 indoor scenes across 6 areas, with each point annotated with one of 13 semantic labels. Data preprocessing follows the method of Qi et al. (2017a), where input features include point coordinates, RGB color, and normalized positions. As per most relevant studies in the literature, area 5 is used as the test set, while the remaining areas are used for training, and the mean IoU (mIoU) is adopted as an evaluation metric.

ScanObjectNN (PB_T50_RS variant) contains 14,450 valid samples across 15 categories, with 11,636 samples for training and 2,814 for testing. Except for using only coordinates as input, data preprocessing and evaluation metric are consistent with those used for ModelNet40 dataset.

### B.2   IMPLEMENTATION

To better understand the model's structure and implementation, we list detailed network architectures and training settings across different datasets in Tab. 11.

## C   MORE ABLATION STUDIES

### C.1   ABLATION COMPARISON ON THE SERIALIZATION

**Serialization**. Based on point cloud serialization, the context-scan state space generates a continuous scanning path with inter-point structural dependencies. To validate the rationale behind selecting

| Configurations | ModelNet40 | ScanObjectNN | ShapeNet | S3DIS |
|---|---|---|---|---|
| Training epochs | 500 | 500 | 600 | 500 |
| Optimizer & Scheduler | Adamw & CosLR | AdamW & CosLR | Adamw & CosLR | Adamw & CosLR |
| Weight decay | 0.01 | 0.01 | 0.01 | 0.01 |
| Learning rate | 8e-4 | 4e-4 | 1e-3 | 1e-3 |
| Warmup epochs | 10 | 20 | 10 | 10 |
| Batch size | 24 | 24 | 24 | 12 |
| Embedding channels | 48 | 48 | 96 | 48 |
| KNN | 8 | 8 | 8 | 8 |
| IPP ratio | 8 | 8 | 8 | 16 |
| Encoder depth | [1, 1, 1, 1] | [1, 1, 1, 1] | [2, 2, 6, 2] | [1, 2, 3, 1] |
| Encoder channels | [48, 96, 192, 384] | [48, 96, 192, 384] | [96, 192, 384, 768] | [96, 192, 384, 768] |
| Decoder depth | - | - | [1, 1, 1, 1] | [1, 1, 1, 1] |
| Decoder channels | - | - | [768, 384, 192, 96] | [768, 384, 192, 96] |
| Downsampling stride | [4, 4, 4] | [4, 4, 4] | [4, 4, 4] | [4, 4, 4] |
| MLP ratio | 4 | 4 | 4 | 4 |
| QKV bias | True | True | True | True |
| Dropout | 0.3 | 0.3 | 0.3 | 0.3 |
| Augmentation | RandomScale RandomShift ShufflePoint | ShufflePoint RandomScale RandomRotate | RandomScale RandomShift ShufflePoint | RandomScale RandomFlip RandomJitter ChromaticAutoContrast ChromaticTranslation ChromaticJitter |

Table 11: Detailed implementation configurations.

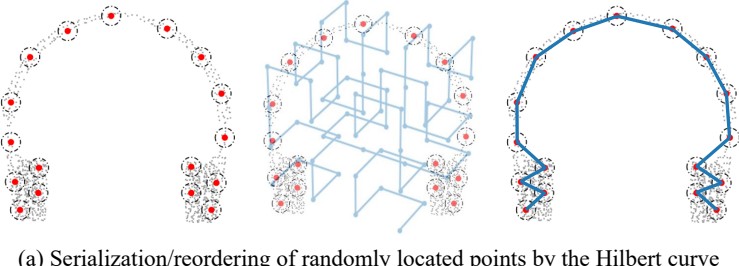

(a) Serialization/reordering of randomly located points by the Hilbert curve

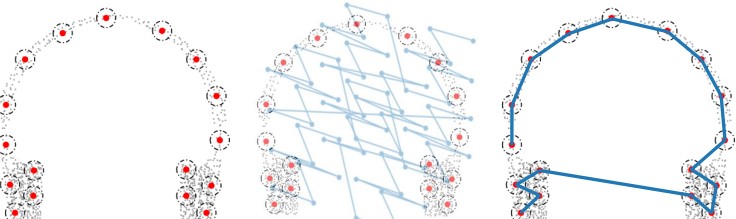

(b) Serialization/reordering of randomly located points by the Z-order curve

Figure 5: Comparison of the serialization of randomly located points by the Hilbert curve (top) and Z-Order curve (bottom).

the Hilbert curve, Tab. 12 compares the ablation results of various serialization strategies. When two serialization methods are employed, they are applied separately to the two directions of the bidirectional S6. Quantitative analysis indicates that while combining multiple serialization strategies can provide richer spatial information, differences in spatial relationships between these strategies may introduce confusion and interfere with the learning of spatial consistency. By leveraging its exceptional spatial locality-preserving property, as shown in Fig. 5, the Hilbert curve establishes a high-fidelity spatially adjacent scanning path with reliable inter-point structural dependencies for the state space model. This aligns with the visual search pattern of eye movements scanning continuously along spatially adjacent regions, thereby achieving an optimal balance between accuracy and efficiency. To further discuss the specific advantages of the Hilbert curve over learnable serial-

| Serialization | Params | FLOPs | Throughput | OA |
|---|---|---|---|---|
| None | 7.36M | 0.610G | 219FPS | 91.34 |
| Hilbert | 7.36M | 0.610G | 163FPS | **94.17** |
| Z-Order | 7.36M | 0.610G | 209FPS | 93.06 |
| Hilbert & Trans-Hilbert | 7.36M | 0.610G | 133FPS | 93.78 |
| Hilbert & Z-Order | 7.36M | 0.610G | 155FPS | 93.52 |
| Learnable Serialization | 8.04M | 0.723G | 168FPS | 92.78 |

Table 12: Ablation results with multiple serialization strategies.

ization strategies in terms of spatial locality preservation, continuity, and computational efficiency, we compare with the learnable serialization strategy from the latest research (Zha et al., 2025). It is intuitively observed that, compared to the learnable serialization, the Hilbert curve exhibits higher computational efficiency and superior spatial locality preservation and continuity. We attribute the poorer performance of the learnable serialization to the fact that it is an adaptive method for determining geometric correlation between points, but this approach possesses much less geometry-specific inductive biases compared to space-filling curves. In summary, the Hilbert curve introduces more precise inductive bias regarding geometric correlation compared to the Z-Order curve and learnable serialization strategies.

## C.2 ABLATION COMPARISON OF IPP AND FPS

In our work, we employ the proposed Induced Point Pooling (IPP) for spatial downsampling. To investigate its ability to flexibly adapt to the non-uniform distribution of point clouds for global semantic integration, we present ablation results comparing IPP with the Farthest Point Sampling (FPS) at different sampling rates in Fig. 6, where /N denotes the sampling rate relative to the input number of points, and None indicates the absence of the spatial downsampling branch. Intuitively, at low sampling rates, both methods exhibit comparable performance, indicating that FPS can obtain better global semantics with its excellent spatial coverage when sufficient sampling points are available. However, as the downsampling rate increases, the performance of FPS declines sharply. At a sampling rate of /256, its accuracy approaches the baseline without the downsampling branch, suggesting its inability to capture critical semantics from non-uniform point clouds at high sparsity. In contrast, IPP exhibits a more gradual performance decrease, maintaining an excellent accuracy of 93.39% even at the /256 sampling rate. This confirms that IPP, through trainable induced points that adaptively learn the point cloud distribution, can more flexibly and robustly integrate global semantics, providing effective coarse-grained information for the point-focused attention.

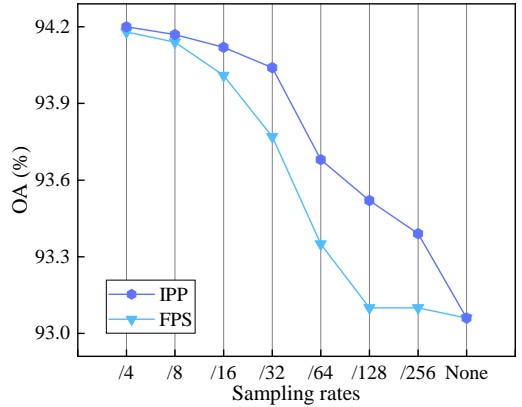

Figure 6: Quantitative results of IPP and FPS.

## D COMPLEXITY ANALYSIS

Following the same settings as Eq.(6) and considering the feature transformation, the computational complexities of each module in the point-focused attention are as follows:

$$\Omega\left(\text{LNB}\right) = 3ND^2 + 2NKD$$
$$\Omega\left(\text{SSD}\right) = ND^2 + 2MD^2 + 2NMD \tag{10}$$
$$\Omega\left(\text{IPP}\right) = 2ND^2 + 2NMD$$

Finally, we have a complexity of $\Omega\left(\text{PFA}\right) = 6ND^2 + 2MD^2 + 2NKD + 4NMD$ for the point-focused attention. Since the context-scan state space inherits the linear complexity inherent in the state space model, the overall network exhibits linear complexity.

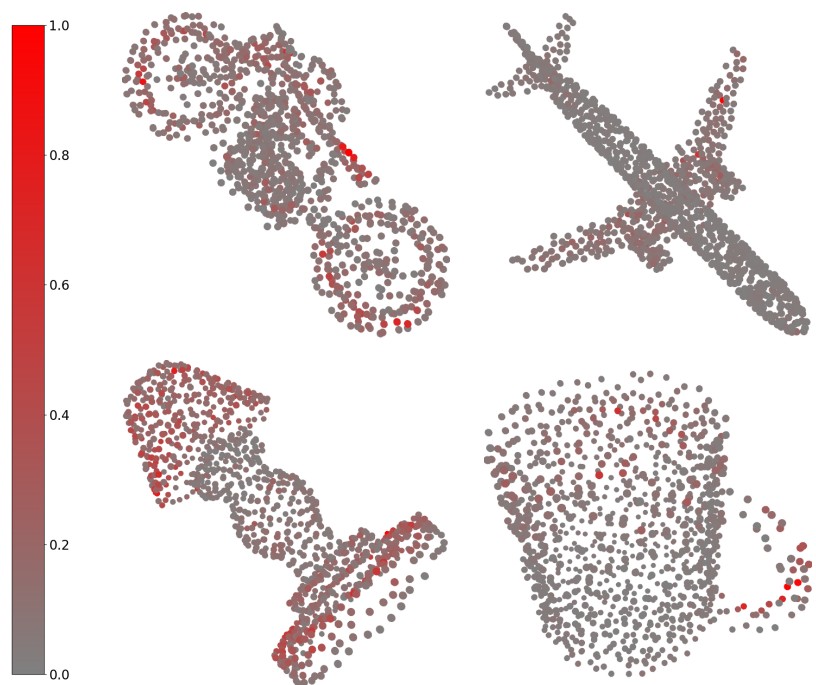

Figure 7: Visualization results of attention heatmaps.

# E VISUALIZATIONS

## E.1 ATTENTION HEATMAPS

To better understand the attention responses and the advantages of the proposed method, we present in Fig. 7 the attention heatmaps of PointLearner for different objects, generated from attention weights in the local neighbor branch of the last point-focused attention layer within the decoder. These attention heatmaps illustrate that, through a fully understanding of the bio-inspired visual perception, our method effectively focuses on critical information for semantic inference to achieve outstanding performance, such as the tires and seat on the motorcycle, as well as the base and cover of the lamp.

## E.2 QUALITATIVE COMPARISON

To intuitively demonstrate the performance of our network, we present in Fig. 8 the visualization results of our network alongside the top-performing SSM method (PointMamba (Liang et al., 2024)) and attention method (GAD (Li et al., 2024b)) from Tab. 2 on ShapeNet dataset, where the red points denote that these points are misclassified. The comparison of the visualization results reveals that our network is able to achieve better part segmentation results at the boundaries of objects.

# F FUTURE WORK

In our experimental comparisons, it is observed that most existing attention models employ pre-training methods to improve performance, with the self-supervised pre-training paradigm dominating. Self-supervised pre-training methods can leverage large amounts of unlabeled data to enhance feature modeling capabilities, as well as help Transformer models with large receptive fields achieve effective local or structural modeling by increasing data scale. Although self-supervised pre-training on large-scale point cloud datasets has been proven effective for improving the accuracy of Transformer models, the compatibility of existing Transformer self-supervised pre-training methods on hybrid architectures, as well as self-supervised pre-training strategies specifically tailored for hybrid architectures, remain underexplored. Hence, it is a promising direction for future research to collect

more data and design self-supervised learning methods for hybrid models, as in PPT (Wu et al., 2024b) and Sonata (Wu et al., 2025) designed for Transformer models.

## G  THE USE OF LARGE LANGUAGE MODELS (LLMs)

The authors utilized large language models (LLMs) to a limited extent for proofreading and improving grammatical correctness. All key aspects of the research, encompassing innovation, conceptual development, and literature discovery, were solely driven by the authors without LLM assistance.

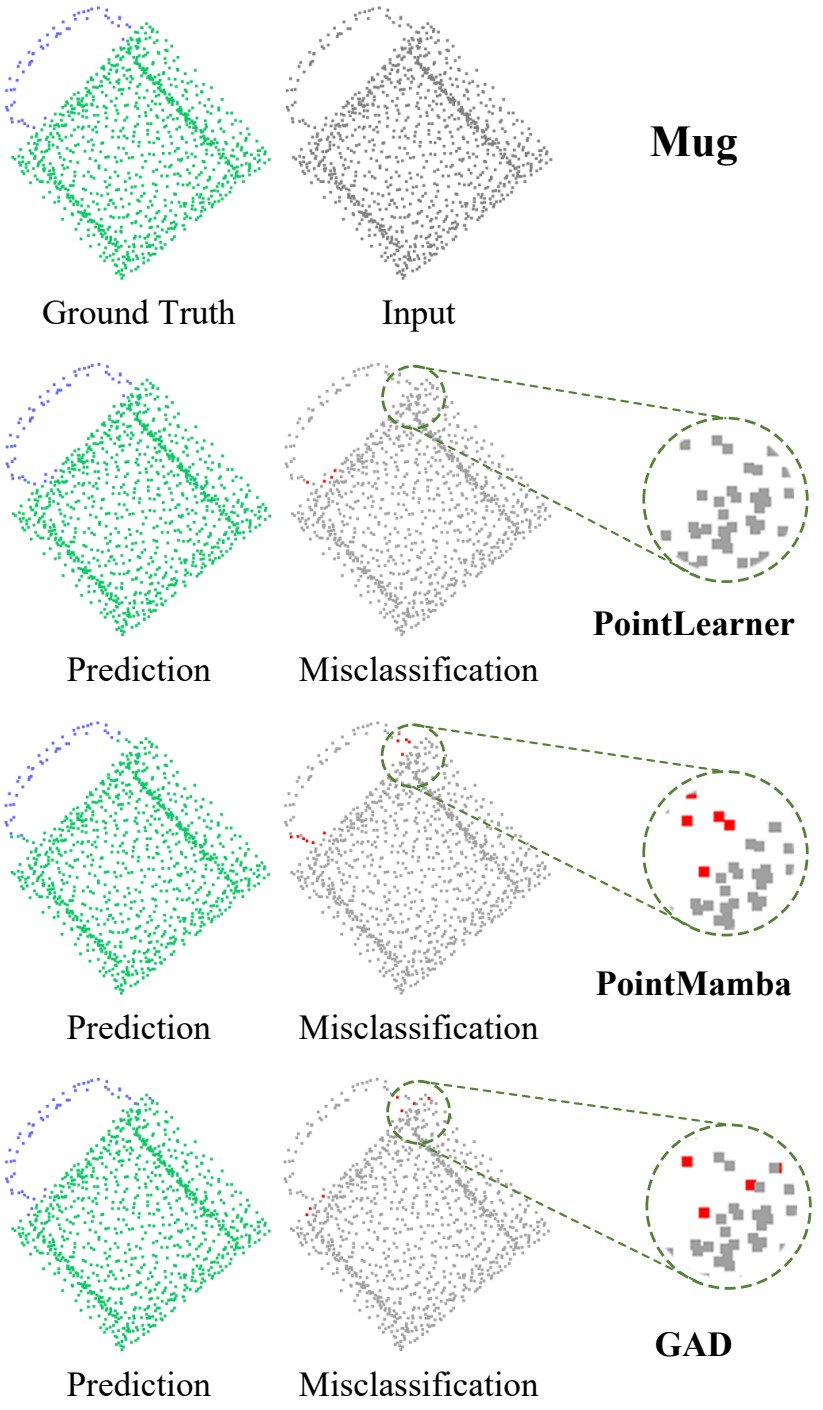

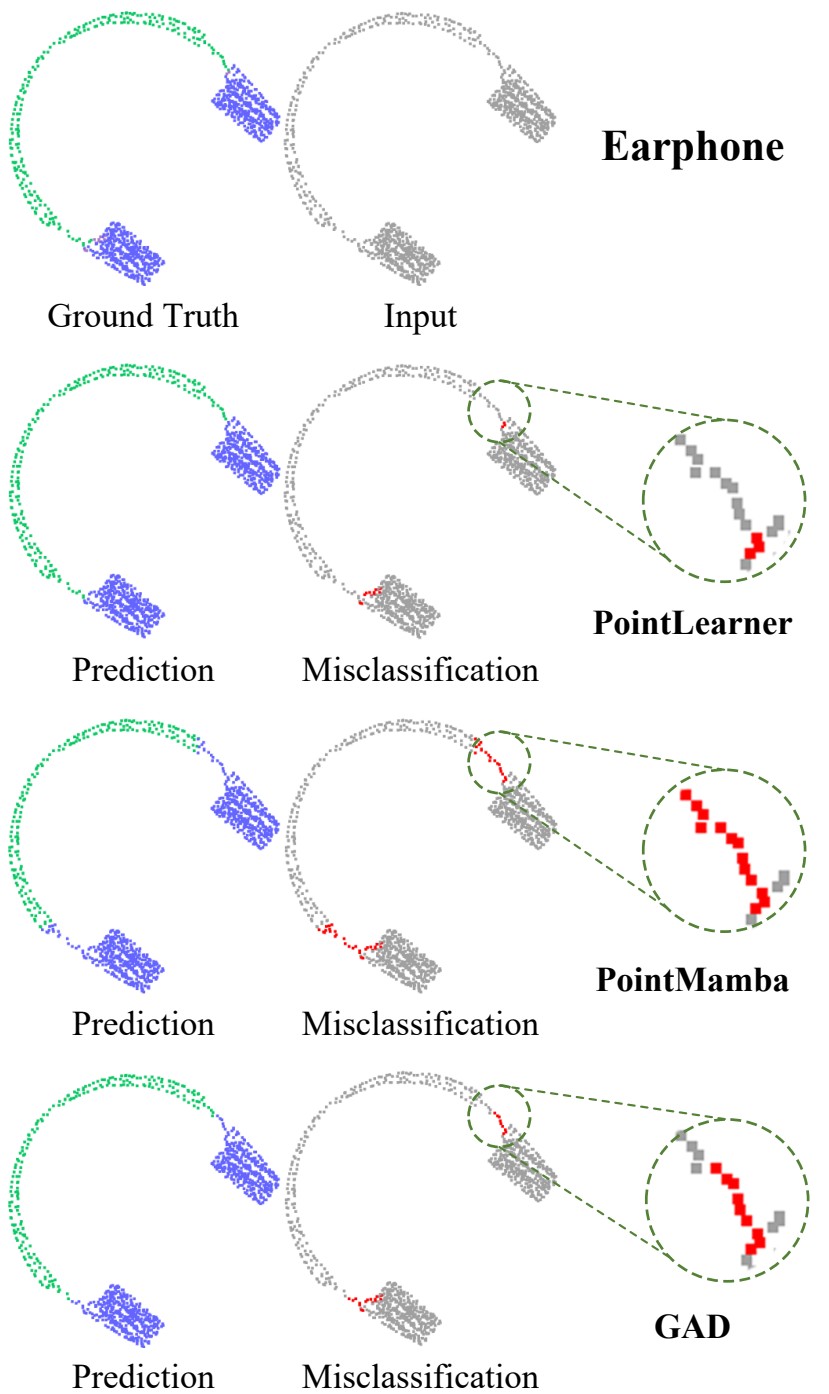

Ground Truth          Input          **Earphone**

Prediction          Misclassification          **PointLearner**

Prediction          Misclassification          **PointMamba**

Prediction          Misclassification          **GAD**

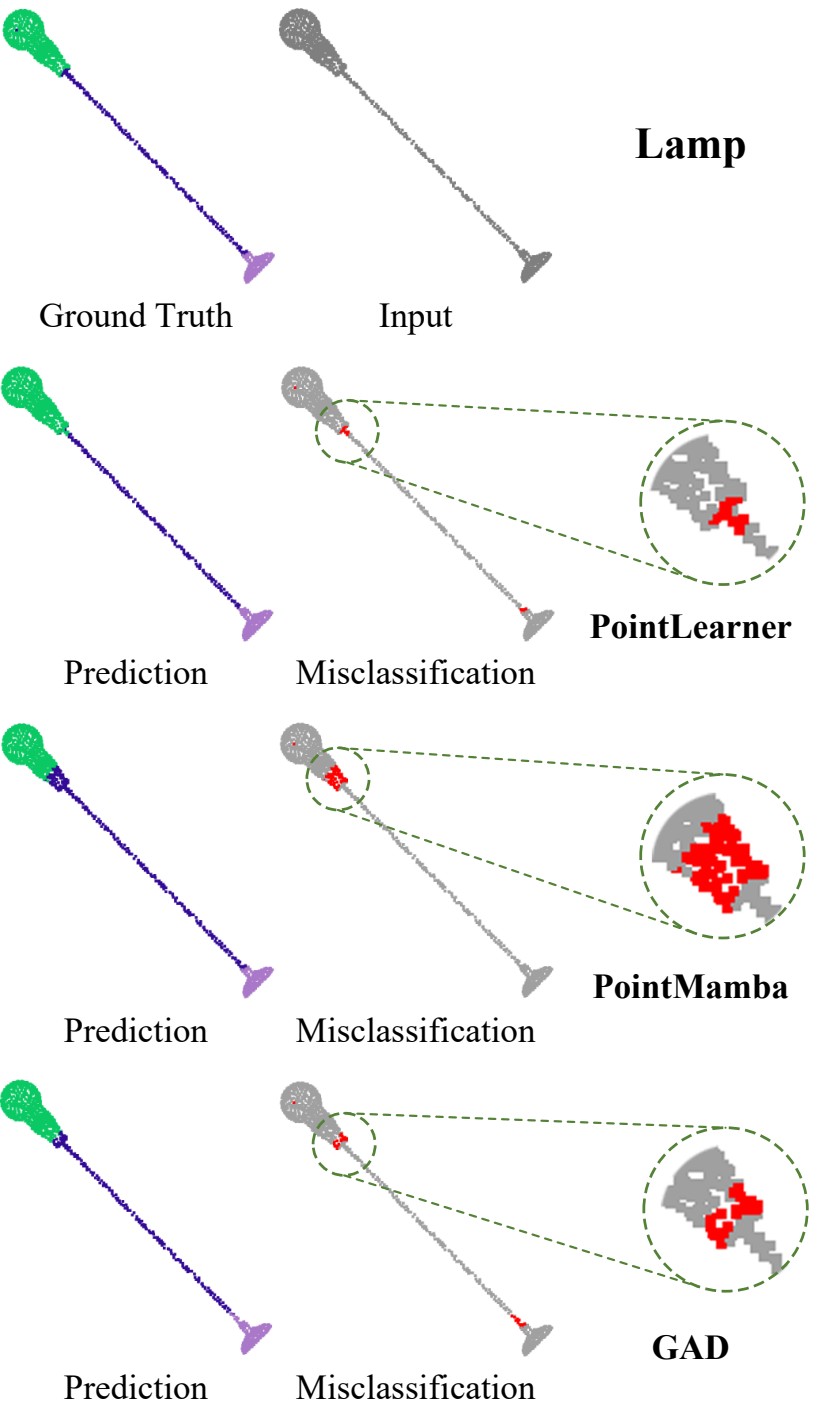

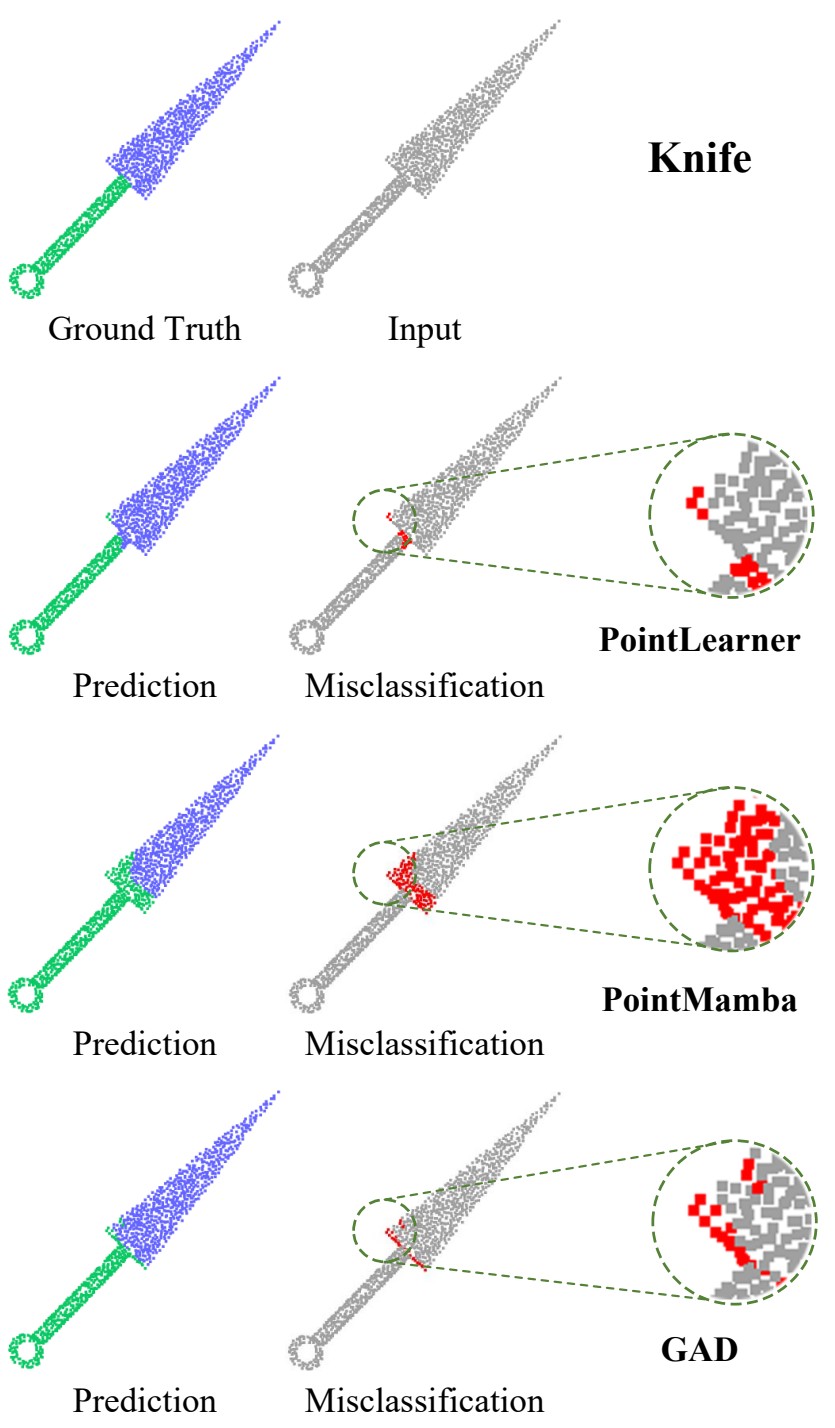

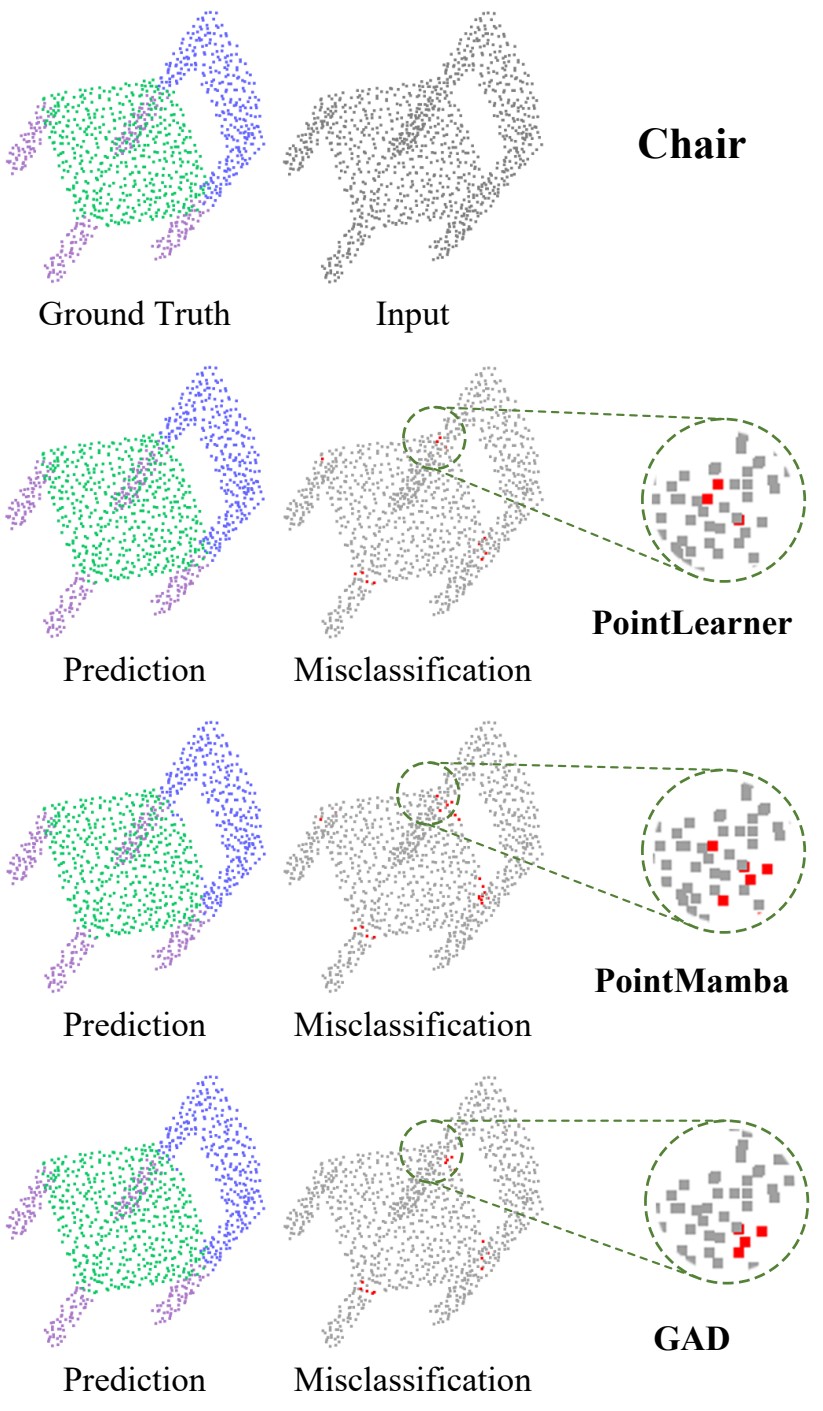

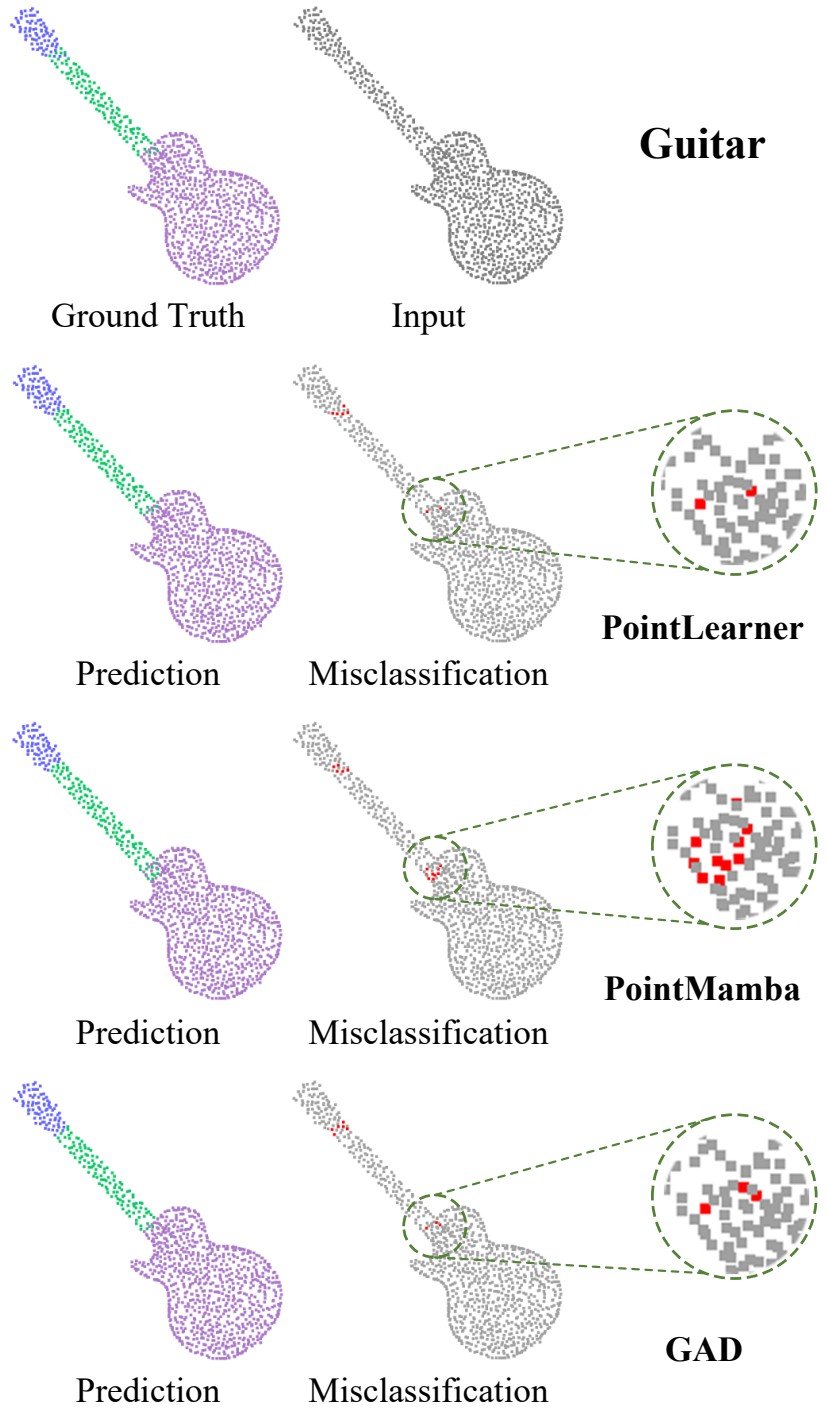

Figure 8: Visualization results of PointLearner, PointMamba, and GAD.

