# OpenReview forum: "Point-Focused Attention Meets Context-Scan State Space: Robust Biological Visual Perception for Point Cloud Representation"
_ICLR.cc/2026/Conference — ICLR 2026 Poster_

### Official Review · Reviewer_Qkeu · 2025-10-26

**Soundness:** 3
**Presentation:** 2
**Contribution:** 2
**Rating:** 6
**Confidence:** 4

**Summary:**

This paper proposes PointLearner, a biomimetic point cloud representation learning network that aims to simulate the "focal-context" perception mechanism of biological visual systems to collaboratively capture both the local fine structure and global context dependencies of point clouds. Specifically, Point-focused attention models local fine details and global semantic dependencies through a local neighbor branch and a spatial downsampling branch. Context-scan state space uses a Hilbert curve to serialize the point cloud and integrates a bidirectional S6 model to dynamically scan and integrate the global semantic structure.

**Strengths:**

1. The dual-path design of local neighbor branch and spatial downsampling branch effectively balances the contribution of local structure and global semantics.
2. To address the non-uniform distribution of point clouds, the paper introduces a learnable inducing points pooling method. By enabling attention interaction between inducing points and the original point cloud, this method flexibly extracts global semantics and shows more stable performance at higher sampling rates compared to the farthest point sampling method.
3. The model demonstrates promising performance improvements over existing methods on four mainstream point cloud datasets (ModelNet40, ShapeNet, S3DIS, ScanObjectNN).

**Weaknesses:**

1. The paper does not provide a comparison of the computational cost of the proposed model with previous methods. Each encoder block not only incorporates a two-branch architecture but also adds a state space model for global feature modeling. This may lead to a significant increase in computational overhead, which requires further clarification.
2. The spatial downsampling branch of the Point-focused attention is capable of capturing global features, so the motivation for introducing the state space model for additional global feature modeling is not well justified. While the authors mention simulating eye jump inference, this design still seems redundant and lacks sufficient novelty, as it closely resembles the design of PointMamba.
3. The details of model training are not sufficiently clear. It is recommended to include a more detailed description of the model architecture across different datasets in the form of tables to help readers better understand the model’s structure and implementation.
4. The experimental comparison is flawed, as it does not distinguish self-supervised models, which could lead to unfair comparisons. For example, pre-trained models on ShapeNet, such as PointGST, may not directly apply to segmentation tasks in indoor scenes. To make the comparison fairer on the S3DIS dataset, it would be more appropriate to include self-supervised methods pre-trained on indoor scene data, such as Sonata.

**Questions:**

1. The serialization strategy in this paper converts 3D data into a 1D sequence, which may disrupt the adjacency relationships between points. Mamba3D argues that serialization is unnecessary, and StructMamba3D also forgoes serialization in favor of spatial state modeling to preserve adjacency relationships. Why does this paper still choose to adopt the Hilbert curve serialization strategy? Are there sufficient experimental results to demonstrate its advantage in terms of model performance?
2. Can the proposed method be applied in the context of self-supervised learning? Leveraging large amounts of unlabeled data to enhance feature modeling capabilities is an important direction. Increasing the dataset size can also help transformer models with large receptive fields to achieve good local or structural modeling. Investigating the impact of self-supervised pre-training on model performance and validating the model’s scalability would further enhance its applicability.

---

> ### Author Response · Authors · 2025-11-20
> **Response to Reviewer Qkeu (Part 1/4)**
>
> **Weakness 1.** The paper does not provide a comparison of the computational cost of the proposed model with previous methods. Each encoder block not only incorporates a two-branch architecture but also adds a state space model for global feature modeling. This may lead to a significant increase in computational overhead, which requires further clarification.
>
> **Response:** Thank you for your insightful feedback. To further analyze the computational overheads of PointLearner, we compared it with several previous state-of-the-art methods on S3DIS dataset, and the params, latency, and memory footprint in a single inference are selected as evaluation metrics. Specifically, to ensure a fair comparison, the latency and memory footprint in a single inference are taken as the average values obtained over the entire S3DIS test set with each scene on the same RTX 4090 GPU. Tab. 5 presents the params, latency, and memory footprint of our network and multiple previous state-of-the-art methods in a single inference. Swin3D is a heavy attention network that, unlike PTv3, does not incorporate efficiency improvements for attention computations. Furthermore, it can be observed that, with a similar number of parameters, PointLearner, as a hybrid network, achieves a superior trade-off between computational overheads and performance compared to the fully optimized PTv3 and the HydraMamba that benefits from S6's excellent properties. We attribute this to the following two aspects: (1) PointLearner is able to achieve powerful local geometric modeling and long-range dependency interactions from the perspective of biological vision through hybrid operators, which simultaneously requires less layer stacking; (2) Although our basic block contains multiple components, they are essentially lightweight with linear complexity, while also benefiting from the hardware-optimized algorithms of FlashAttention.
>
> | Networks | Operator | Params | Latency | Memory | mIoU |
> | :--- | :--- | :--- | :--- | :--- | :--- |
> | HydraMamba (Qu et al., 2025) | SSM | 63.14M | 54ms | **5.9G** | 73.6 |
> | PTv3 (Wu et al., 2024a) | Attention | 46.17M | **49ms** | 6.3G | 73.4 |
> | Swin3D 3 (Yang et al., 2025) | Attention | 71.15M | 365ms | 10.7G | 72.5 |
> | PointLearner | Hybrid | 52.78M | 63ms | 6.5G | **74.3** |
>
> **Table 5: Params, latency, and memory footprint of our network and previous state-of-the-art methods in a single inference on S3DIS dataset.**

---

> ### Author Response · Authors · 2025-11-20
> **Response to Reviewer Qkeu (Part 2/4)**
>
> **Weakness 2.** The spatial downsampling branch of the Point-focused attention is capable of capturing global features, so the motivation for introducing the state space model for additional global feature modeling is not well justified. While the authors mention simulating eye jump inference, this design still seems redundant and lacks sufficient novelty, as it closely resembles the design of PointMamba.
>
> **Response:** We sincerely appreciate this highly insightful comment.
>
> First, let us discuss the designed modules beyond the framework of biological vision. The downsampling branch in the point-focused attention provides a coarse-grained perception of global features, which, as shown in Tab. 10, is factually insufficient for a full understanding of global semantics. However, the quadratic complexity of the attention mechanism makes fine-grained global perception infeasible. Therefore, we further introduce the State Space Model (SSM) for additional coarse-grained global feature modeling. By coordinating SSM with the downsampling branch, our model approximates fine-grained global perception with linear complexity, thereby avoiding quadratic complexity. **Notably**, we consider the State Space Model to also be a coarse-grained perception for point clouds. This is because SSM involves a recurrent inference process, and the selective mechanism of S6 is prone to losing fine-grained features during this recurrence, achieving a coarse-grained perception similar to the downsampling branch.
>
> Then, we consider these two global perception modules with linear complexity within the framework of biological vision. Functionally, the modules help the visual focus achieve fine-grained global perception with linear complexity, aligning with the utilization of finite neural resources in biological vision [1].
>
> Finally, our context-scan state space is generally different from PointMamba. In adapting S6's causal nature, both consider using the Hilbert curve to establish inter-point structural dependencies in point sequences for geometric inference. While, in adapting S6's unidirectional modeling, PointMamba achieves global modeling by first concatenating the serialization results of the Hilbert and Trans-Hilbert curves and then using S6 only once. This practice may lead to confusion due to multiple geometry-specific inductive biases introduced by the spatial relationships of different serialization results [2] (as shown in Tab. 12), and it also contradicts the operational logic of biological vision. In our context-scan state space, the bidirectional S6 achieves global perception over one geometry-specific inductive bias, thereby avoiding confusion, and also effectively simulates the eye's back-and-forth saccade to enhance inference ability in complex scenes.
>
> [1] B. A. Wandell. Foundations of Vision. Sinauer Associates, 1995.
>
> [2] N. Köprücü, D. Okpekpe, and A. Orvieto. Nimba: Towards robust and principled processing of point clouds with ssms. arXiv: 2411.00151, 2024.
>
> | PFA | CSSS | Params | FLOPs | Throughput | OA |
> | :--- | :--- | :--- | :--- | :--- | :--- |
> | ✓ | × | 4.57M | 0.487G | 198FPS | 92.93 |
> | × | ✓ | 5.37M | 0.463G | 231FPS | 91.94 |
> | ✓ | ✓ | 7.36M | 0.610G | 163FPS | **94.17** |
>
> **Table 10: Ablation results with PFA and CSCC.**
>
> | Serialization | Params | FLOPs | Throughput | OA |
> | :--- | :--- | :--- | :--- | :--- |
> | None | 7.36M | 0.610G | 219FPS | 91.34 |
> | Hilbert | 7.36M | 0.610G | 163FPS | **94.17** |
> | Z-Order | 7.36M | 0.610G | 209FPS | 93.06 |
> | Hilbert & Trans-Hilbert | 7.36M | 0.610G | 133FPS | 93.78 |
> | Hilbert & Z-Order | 7.36M | 0.610G | 155FPS | 93.52 |
> | Learnable Serialization | 8.04M | 0.723G | 168FPS | 92.78 |
>
> **Table 12: Ablation results with multiple serialization strategies.**
>
> **Weakness 3.** The details of model training are not sufficiently clear. It is recommended to include a more detailed description of the model architecture across different datasets in the form of tables to help readers better understand the model’s structure and implementation.
>
> **Response:** Following your comment, to better understand the model's structure and implementation, we have listed detailed network architectures and training settings across different datasets in a tabular form in Appendix B.2.

---

> ### Author Response · Authors · 2025-11-20
> **Response to Reviewer Qkeu (Part 3/4)**
>
> **Weakness 4.** he experimental comparison is flawed, as it does not distinguish self-supervised models, which could lead to unfair comparisons. For example, pre-trained models on ShapeNet, such as PointGST, may not directly apply to segmentation tasks in indoor scenes.  To make the comparison fairer on the S3DIS dataset, it would be more appropriate to include self-supervised methods pre-trained on indoor scene data, such as Sonata.
>
> **Response:** We sincerely appreciate your meticulous review.
>
> To standardize experimental comparisons, we have marked methods utilizing pre-training strategies with the symbol “†”.
>
> In our experimental comparisons, most existing attention models employ pre-training methods to improve performance, with the self-supervised pre-training paradigm dominating. These models are typically pre-trained on ShapeNet dataset and then fine-tuned for various downstream tasks. By learning generic features of point clouds, such as geometric details and spatial structures, via a self-supervised method prior to fine-tuning downstream tasks, these attention models should have achieved better performance due to learning on ShapeNet and the target task dataset. Despite this, our method still achieves superior performance compared to these self-supervised pre-trained Transformer models.
>
> Regarding to PointGST, it learns general features of point clouds, such as geometric details and spatial structures, via a self-supervised method on ShapeNet, and these learned features could possibly be applied to downstream segmentation tasks in indoor scenes. Pretraing on ShapeNet and finetuning on S3DIS dataset is also the way adopted in the original PointGST paper.
>
> Sonata achieves excellent results by a self-supervised pre-training method tailored for Transformer model on large-scale indoor scene data, with mIoU of 72.3 at linear probing, 74.5 at decoder probing, and 76.0 at full fine-tuning. When comparing the segmentation results, the former case is inferior to our result, and the latter two cases are slightly better than ours. However, directly comparing our method with self-supervised pre-training methods is somewhat unfair, unless when a self-supervised pre-training technique tailored for hybrid models is used to pre-train our method on the same large-scale indoor scene data collected by Sonata [1]. Therefore, the discussion regarding collecting more data and designing self-supervised learning methods for hybrid models like ours, as in PPT [2] and Sonata [1] designed for Transformer models, is expanded upon in Appendix F.
>
> [1] X. Y. Wu, D. DeTone, D. Frost, T. W. Shen, C. Xie, N. Yang, J. Engel, R. Newcombe, H. S. Zhao, and J. Straub. Sonata: Self-supervised learning of reliable point representations. In Proc. IEEE/CVF Conference on Computer Vision and Pattern Recognition (CVPR), pp. 22193–22204, Nashville, TN, USA, Jun. 2025.
>
> [2] X. Y. Wu, Z. T. Tian, X. Wen, B. H. Peng, X. H. Liu, K. C. Yu, and H. S. Zhao. Towards large-scale 3d representation learning with multi-dataset point prompt training. In Proc. IEEE/CVF Conference on Computer Vision and Pattern Recognition (CVPR), pp. 19551–19562, Seattle, WA, USA, Jun. 2024b.
>
> **Appendix F**
>
> In our experimental comparisons, it is observed that most existing attention models employ pre-training methods to improve performance, with the self-supervised pre-training paradigm dominating. Self-supervised pre-training methods can leverage large amounts of unlabeled data to enhance feature modeling capabilities, as well as help Transformer models with large receptive fields achieve effective local or structural modeling by increasing data scale. Although self-supervised pre-training on large-scale point cloud datasets has been proven effective for improving the accuracy of Transformer models, the compatibility of existing Transformer self-supervised pre-training methods on hybrid architectures, as well as self-supervised pre-training strategies specifically tailored for hybrid architectures, remain underexplored. Hence, it is a promising direction for future research to collect more data and design self-supervised learning methods for hybrid models, as in PPT (Wu et al., 2024b) and Sonata (Wu et al., 2025) designed for Transformer models.

---

> ### Author Response · Authors · 2025-11-20
> **Response to Reviewer Qkeu (Part 4/4)**
>
> **Question 1.** The serialization strategy in this paper converts 3D data into a 1D sequence, which may disrupt the adjacency relationships between points. Mamba3D argues that serialization is unnecessary, and StructMamba3D also forgoes serialization in favor of spatial state modeling to preserve adjacency relationships. Why does this paper still choose to adopt the Hilbert curve serialization strategy? Are there sufficient experimental results to demonstrate its advantage in terms of model performance?
>
> **Response:** Thank you for your valuable comment.
>
> As an LSTM-like operator, SSM involves a permutation-variant recurrent inference process. Therefore, when SSM is used as token mixer for unstructured point sets, this necessarily requires geometric relationships in point sequences. While the Hilbert curve may disrupt certain adjacency relationships between points, it is currently the method that provides the best geometry-specific inductive bias according to the experimental results of different serialization strategies in Tab. 12. Importantly, the Hilbert curve serialization strategy can be used to simulate the eye’s saccade process. By integrating with the point-focused attention, we arrive at our final model, i.e., a biological inspired focus-then-context pipeline.
>
> In Mamba3D, SSM is used as an improvement on channel mixer, replacing MLP and operating on channels. Unlike the unstructured nature between point tokens in a point set, the structural relationships learned by a point are contained between the channels of each point token, which is used for inference to achieve channel fusion. This is why SSM in Mamba3D does not require serialization. We believe that the Local Norm Pooling (LNP), as token mixer in Mamba3D, is key to capturing geometric semantics through inter-point relationships, but it is achieved directly through spatial associations (KNN) and is unrelated to SSM. Therefore, this component also does not require serialization. StructMamba3D forgoes the serialization mainly through designing spatial states and using them as proxies to preserve spatial dependencies among points.
>
> **Question 2.** Can the proposed method be applied in the context of self-supervised learning? Leveraging large amounts of unlabeled data to enhance feature modeling capabilities is an important direction. Increasing the dataset size can also help transformer models with large receptive fields to achieve good local or structural modeling. Investigating the impact of self-supervised pre-training on model performance and validating the model’s scalability would further enhance its applicability.
>
> **Response:** Thank you for your thought-provoking question. We understand the advantages of the self-supervised paradigm and believe the proposed method can be applied in the context of self-supervised learning. However, as demonstrated by the comparative methods, most current self-supervised pre-training methods are tailored for pure Transformer models. Their compatibility with SSM-based architectures, as well as self-supervised pre-training strategies specifically designed for SSM architectures, remains underexplored—let alone hybrid architectures. Therefore, investigating the compatibility of existing Transformer self-supervised pre-training methods with hybrid models, or developing novel self-supervised pre-training strategies specifically for hybrid models, is a promising direction for future research. We have added this perspective to Appendix F.

---

> ### Author Response · Authors · 2025-11-27
>
> Dear Reviewer Qkeu,
>
> Thank you very much for your positive rating on our submission. We genuinely appreciate the time and effort you dedicated to reviewing our work. We hope that our rebuttal has successfully addressed your concerns and clarified any questions you had regarding the paper. As the discussion phase is ongoing, please feel free to raise any further questions you may have about our responses or the main text of the paper. We look forward to receiving your feedback.
>
> Best regards,
>
> Paper 21998 Authors

---

### Official Review · Reviewer_i3Jg · 2025-10-28

**Soundness:** 3
**Presentation:** 3
**Contribution:** 3
**Rating:** 8
**Confidence:** 3

**Summary:**

This paper proposes PointLearner, a point cloud representation learning network that is inspired by biological vision mechanisms.
The method introduces two main components: (1) a point-focused attention module that mimics foveal vision by combining fine-grained local neighbor attention with coarse-grained global semantics from a novel induced point pooling strategy, and (2) a context-scan state space module that serializes point clouds using a Hilbert curve and applies bidirectional S6 to simulate saccadic eye movements for global scene inference.
The authors demonstrate state-of-the-art performance on ModelNet40, ShapeNet, S3DIS, and ScanObjectNN datasets, with particular robustness to strong noise and varying point densities.

**Strengths:**

1. The analogy to foveal vision and saccadic eye movements is well-motivated, intuitively aligning with the challenges of point cloud learning, including local detail preservation and global context integration.

2. The work effectively combines the strengths of attention (local geometry sensitivity) and state space models (long-range dependency modeling) within a biologically plausible framework, demonstrating the potential of such hybrid paradigms.

**Weaknesses:**

1. The biological analogy is intuitive but somewhat heuristic.
A more rigorous theoretical or empirical analysis of how closely the modules mimic biological vision would strengthen the motivation and be helpful for understanding.

2. Visualizing heat maps could be useful for better understanding the attention responses and the advantage of the proposed method.

**Questions:**

What are the specific advantages of using the Hilbert curve over the Morton curve or learned serialization strategies in terms of spatial locality preservation, continuity, and computational efficiency?

---

> ### Author Response · Authors · 2025-11-20
> **Response to Reviewer i3Jg (Part 1/1)**
>
> **Weakness 1.** The biological analogy is intuitive but somewhat heuristic. A more rigorous theoretical or empirical analysis of how closely the modules mimic biological vision would strengthen the motivation and be helpful for understanding.
>
> **Response:** Thank you for your insightful comment. In fact, rigorous theoretical or experimental analysis of how closely the modules mimic biological vision is generally difficult to achieve, because it is currently impossible to quantitatively compare our modules' internal representations (or neural activation patterns) with the representations of existing biological visual computation models when processing specific datasets. Therefore, we roughly assess how closely the modules mimic biological vision by the accuracy difference between ablation experiments. For example, in the ablation study in Tab. 8, our proposed competitive normalized fusion aligns with deep dynamic interactions between local fine-grained features and global coarse-grained semantics inherent in foveal vision perception, yielding better accuracy than the additive fusion.
>
> **Weakness 2.** Visualizing heat maps could be useful for better understanding the attention responses and the advantage of the proposed method.
>
> **Response:** Thank you for your valuable comment on the visualization analysis. We fully agree that visualizing attention heatmaps can better understand the attention response and the advantage of the proposed method. Therefore, in Fig. 7 in Appendix E.1, we have added attention heatmaps for PointLearner, generated from attention weights in the local neighbor branch of the last point-focused attention layer within the decoder. These attention heatmaps illustrate that, through a fully understanding of the bio-inspired visual perception, our method effectively focuses on critical information for semantic inference to achieve outstanding performance, such as the tires and seat on the motorcycle, as well as the base and cover of the lamp.
>
> **Question 1.** What are the specific advantages of using the Hilbert curve over the Morton curve or learned serialization strategies in terms of spatial locality preservation, continuity, and computational efficiency?
>
> **Response:** The Morton curve is a type of space-filling curves, also commonly referred to as the Z-Order curve or Lebesgue curve. The specific advantages of the Hilbert curve over the Z-Order curve in terms of spatial locality preservation, continuity, and computational efficiency have been extensively discussed in Tab. 12 in Appendix C.1. Specifically, the Z-Order curve exhibits higher computational efficiency (same parameters and FLOPs but higher throughput), while the Hilbert curve offers superior spatial locality preservation and continuity (higher accuracy) – an insight consistent with prior research [1].
>
> To further discuss the specific advantages of the Hilbert curve over learnable serialization strategies in terms of spatial locality preservation, continuity, and computational efficiency, we have added the learnable serialization strategy from the latest research [2] to Tab. 12. It is intuitively observed that, compared to the learnable serialization, the Hilbert curve exhibits higher computational efficiency (lower parameters and FLOPs, and comparable throughput) and superior spatial locality preservation and continuity (higher accuracy). We attribute the poorer performance of the learnable serialization to the fact that it is an adaptive method for determining geometric correlation between points, but this approach possesses much less geometry-specific inductive biases compared to space-filling curves. In summary, the Hilbert curve introduces more precise inductive bias regarding geometric correlation compared to the Z-Order curve and learnable serialization strategies.
>
> [1] A. Nordin and A. Telles. Comparing the locality preservation of z-order curves and hilbert curves. 2023.
>
> [2] Y. H. Zha, Y. Z. Wang, H. Guo, J. P. Wang, T. Dai, B. Chen, Z. H. Ouyang, Y. R. Xue, K. Chen, and S. T. Xia. Pma: Towards parameter-efficient point cloud understanding via point mamba adapter. In Proc. IEEE/CVF Conference on Computer Vision and Pattern Recognition (CVPR), pp. 16976–16986, Nashville, TN, USA, Jun. 2025.
>
> | Serialization | Params | FLOPs | Throughput | OA |
> | :--- | :--- | :--- | :--- | :--- |
> | None | 7.36M | 0.610G | 219FPS | 91.34 |
> | Hilbert | 7.36M | 0.610G | 163FPS | **94.17** |
> | Z-Order | 7.36M | 0.610G | 209FPS | 93.06 |
> | Hilbert & Trans-Hilbert | 7.36M | 0.610G | 133FPS | 93.78 |
> | Hilbert & Z-Order | 7.36M | 0.610G | 155FPS | 93.52 |
> | Learnable Serialization | 8.04M | 0.723G | 168FPS | 92.78 |
>
> **Table 12: Ablation results with multiple serialization strategies.**

---

> > ### Comment · Reviewer_i3Jg · 2025-11-25
> >
> > Thanks for the rebuttal. I believe the authors have addressed my concerns, and I would like to maintain my original rating. I will also wait for the other reviewers’ feedback before any further discussion.

---

> > > ### Author Response · Authors · 2025-11-25
> > >
> > > Dear reviewer i3Jg,
> > >
> > > Thank you for your recognition of our responses and for maintaining a positive rating. If you have any further questions during the next discussion period, please let us know, and we would be happy to answer them. Thank you once again!
> > >
> > > Sincerely,
> > >
> > > Paper 21998 Authors

---

### Official Review · Reviewer_5iH6 · 2025-10-31

**Soundness:** 2
**Presentation:** 3
**Contribution:** 3
**Rating:** 6
**Confidence:** 3

**Summary:**

This paper proposes PointLearner, a biologically inspired point cloud learning framework that integrates local geometric modeling and global contextual reasoning. The key idea is to mimic human visual perception, which combines a point-focused attention mechanism  with a context-scan state space. Experiments on several benchmarks show state-of-the-art performance and strong robustness to noise and varying point densities.

**Strengths:**

1. A novel biomimetic perspective. The motivation of this paper is very interesting, which is derived from biological visual analogies, effectively balancing local attention and the understanding of global features.

2. Robust architectural design. The proposed point-focused attention module and state space module are well integrated and can effectively handle noise and sparse sampling, demonstrating good generalization ability.

3. Clear structure and writing. This paper is logically clear and easy to follow.

**Weaknesses:**

Lack of domain-specific motivation. Although the biological vision analogy is creative, the paper does not clearly explain why this mechanism is particularly suitable for point clouds. The proposed “foveation + saccade” process seems domain-agnostic and could also be applied to images or videos. Without a clear justification, the choice of point cloud data feels somewhat arbitrary.

**Questions:**

Why did the authors choose to apply this biomimetic architecture specifically to point clouds rather than to 2D images or videos, where visual perception analogies might seem even more natural?

Have the authors considered testing or conceptually discussing how this framework would perform on other vision data (e.g., images or video) ?

---

> ### Author Response · Authors · 2025-11-20
> **Response to Reviewer 5iH6 (Part 1/1)**
>
> **Weakness 1.** Lack of domain-specific motivation. Although the biological vision analogy is creative, the paper does not clearly explain why this mechanism is particularly suitable for point clouds. The proposed “foveation + saccade” process seems domain-agnostic and could also be applied to images or videos. Without a clear justification, the choice of point cloud data feels somewhat arbitrary.
>
> **Response:** Thank you for your insightful comment. Biological vision possesses powerful perceptual capability, enabling synergistic modeling of local geometries and long-range dependencies to overcome the critical challenge in point cloud representation learning.
>
> In fact, the biological vision analogy can be understood as a framework requiring a "foveation + saccade" process, which can be applied to various data types (just as the visual system can observe everything), including images, videos, and point clouds, while the domain-specific motivation is reflected in how to implement the "foveation + saccade" process. For example, for foveal visual perception on images, we can respectively use grid convolution and average pooling to achieve high acuity at the focus and low acuity in the periphery. However, due to the domain properties of point clouds, namely their non-uniformity and unstructuredness, this approach is evidently unsuitable. Therefore, the attention mechanism with permutation-invariance and the induced point pooling that is more adaptable to non-uniform distribution are specifically applied to foveal visual perception in the point cloud domain.
>
> **Question 1.** Why did the authors choose to apply this biomimetic architecture specifically to point clouds rather than to 2D images or videos, where visual perception analogies might seem even more natural? Have the authors considered testing or conceptually discussing how this framework would perform on other vision data (e.g., images or video)?
>
> **Response:** Thank you for your valuable comment. As with the response to Weakness 1, our specific biomimetic architecture is constructed guided by the biological vision framework, aligning with the domain-specific motivations of point clouds. Specifically, the point-focused attention adopts the attention mechanism and induced point pooling to adapt to the disorder and non-uniform distribution of point clouds, respectively; while the context-scan state space uses the serialization and state space model to adapt to the sparsity and large number of points in point clouds, respectively.
>
> Our biomimetic architecture does not seem to be directly adaptable to images or videos, because the K-nearest neighbor-based local neighborhood branch and the Hilbert curve do not appear to be very suitable for these grid-structured data. Of course, guided by the biological vision framework, one could also design a domain-specific biomimetic architecture tailored for images or videos—which is exactly what our work aims to convey! As mentioned earlier, for foveal visual perception on images, we can respectively use grid convolution and average pooling to achieve high acuity at the focus and low acuity in the periphery. And, for the eye’s saccade inference, we may use the bidirectional S6 for sequential scanning on pixel patches.

---

> ### Author Response · Authors · 2025-11-27
> **We are open to any further discussion.**
>
> Dear Reviewer 5iH6,
>
> Thank you very much for your positive rating on our submission. We genuinely appreciate the time and effort you dedicated to reviewing our work. We hope that our rebuttal has successfully addressed your concerns and clarified any questions you had regarding the paper. As the discussion phase is ongoing, please feel free to raise any further questions you may have about our responses or the main text of the paper. We look forward to receiving your feedback.
>
> Best regards,
>
> Paper 21998 Authors

---

### Official Review · Reviewer_9P6s · 2025-10-31

**Soundness:** 1
**Presentation:** 3
**Contribution:** 2
**Rating:** 4
**Confidence:** 4

**Summary:**

This paper proposes PointLearner, a point cloud representation learning network that closely aligns with biological vision by employing an active, foveation-inspired processing strategy, thus enabling simultaneous local geometric modeling and long-range dependency interactions.

**Strengths:**

1.The motivation of explaining the network design from a biological vision perspective is interesting;
2.The argument in the Introduction regarding the limitations of local attention is correct;
3.The goal of "synergistically capturing local fine-grained structures and global contextual dependencies" is valid;
4.The experimental results demonstrate the effectiveness of the proposed network architecture;
5.The paper is well-written and easy to understand.

**Weaknesses:**

1.Although the motivation and starting point of the paper are sound, the core ideas of the proposed solution lack novelty;
2.The Fine-grained Perception module, which updates the center token using k neighboring tokens, is very common apart from the novel use of attention;
3.The Coarse-grained Awareness module is essentially a standard approach that uses learnable queries with cross-attention to aggregate features;
4.The Competitive Normalized Fusion is a commonly adopted strategy when a network has multiple parallel branches;
5.Most of the datasets used are pure point cloud data. To better align with the biological vision motivation, it is recommended to include more datasets with RGB information.

**Questions:**

see weakness

---

> ### Author Response · Authors · 2025-11-20
> **Response to Reviewer 9P6s (Part 1/2)**
>
> **Weakness 2**. The Fine-grained Perception module, which updates the center token using k neighboring tokens, is very common apart from the novel use of attention.
>
> **Response:** Thank you for your comment. Due to the domain-specific properties of point clouds, namely, non-uniform distribution and unstructuredness, updating the center token using k neighboring tokens has become a standard paradigm. While, how to aggregate the neighboring tokens is the crucial part of this module. Existing works uses relative distance weighted function [1-7], the octree convolution [8], or local norm pooling [9], etc. Nevertheless, our work updates the center token in the following way. Firstly, we calculate the attention weight between the center token and the neighboring tokens. Then, we update the attention weight by computing the attention weights of both downsampling and neighboring tokens within a single softmax calculation. Finally, we update the center token via the reined attention weights and neighboring token values.
> In summary, we update the center token through deep dynamic interactions between local fine-grained features and global coarse-grained semantics in the competitive normalized mechanism.
>
> [1] Y. Q. Yang, Y. X. Guo, J. Y. Xiong, Y. Liu, H. Pan, P. S. Wang, X. Tong, and B. N. Guo. Swin3d: A pretrained transformer backbone for 3d indoor scene understanding. Computational Visual Media (CVM), 11(1):83–101, Feb. 2025.
>
> [2] Z. H. Li, P. Gao, K. You, C. Yan, and M. Paul. Global attention-guided dual-domain point cloud feature learning for classification and segmentation. IEEE Transactions on Artificial Intelligence (TAI), 5(10):5167–5167, Oct. 2024b.
>
> [3] H. Thomas, Y. H. Tsai, T. D. Barfoot, and J. Zhang. Kpconvx: Modernizing kernel point convolution with kernel attention. In Proc. IEEE/CVF Conference on Computer Vision and Pattern Recognition (CVPR), pp. 5525–5535, Seattle, WA, USA, Jun. 2024.
>
> [4] Y. H. Liu, B. Tian, Y. S. Lv, L. X. Li, and F. Y. Wang. Point cloud classification using content-based transformer via clustering in feature space. IEEE/CAA Journal of Automatica Sinica (JAS), 11(1): 231–239, Jan. 2024b.
>
> [5] J. Park, S. Lee, S. Kim, Y. Y. Xiong, and H. J. Kim. Self-positioning point-based transformer  for point cloud understanding. In Proc. IEEE/CVF Conference on Computer Vision and Pattern Recognition (CVPR), pp. 21814–21823, Vancouver, BC, Canada, Jun. 2023.
>
> [6] Y. B. Gao, X. B. Liu, J. Li, Z. J. Fang, X. Y. Jiang, and K. M. S. Huq. Lft-net: Local feature transformer network for point clouds analysis. IEEE Transactions on Intelligent Transportation Systems (TITS), 24(2):2158–2168, Feb. 2023.
>
> [7] P. Xiang, X. Wen, Y. S. Liu, H. Zhang, Y. Fang, and Z. Z. Han. Retro-fpn: Retrospective feature pyramid network for point cloud semantic segmentation. In Proc. IEEE/CVF International Conference on Computer Vision (ICCV), pp. 17826–17838, Paris, France, Oct. 2023.
>
> [8] P. S. Wang. Octformer: Octree-based transformers for 3d point clouds. ACM Transactions on Graphics (TOG), 42(4):155, Jul. 2023.
>
> [9] X. Han, Y. Tang, Z. X. Wang, and X. Z. Li. Mamba3d: Enhancing local features for 3d point cloud analysis via state space model. In Proc. ACM International Conference on Multimedia (MM), pp. 4995–5004, Melbourne, Australia, Oct. 2024.
>
> **Weakness 3**. The Coarse-grained Awareness module is essentially a standard approach that uses learnable queries with cross-attention to aggregate features.
>
> **Response:** Thank you for your feedback. The coarse-grained awareness requires downsampling and feature aggregation. Due to the disorder and unstructuredness of point clouds, the most commonly adopted method for the coarse-grained awareness is the Farthest Point Sampling (FPS), but we found that FPS does not adapt well to the non-uniform distribution of point clouds, i.e., as the downsampling rate increases, the performance of FPS declines sharply, as illustrated in Fig. 6 in Appendix C.2. To this end, we achieve downsampling while aggregating features, through the learnable queries with cross-attention. Since the size of learnable queries is controllable and they can interact directly with point clouds, our model can flexibly adapt to the non-uniform distribution of point clouds. To the best of our knowledge, designing learnable queries for point downsampling and feature aggregation is **the first attempt** in point cloud representation learning. In addition, the experiment in Appendix C.2 validates the feasibility of learnable queries for the downsampling and feature aggregation of point clouds.

---

> ### Author Response · Authors · 2025-11-20
> **Response to Reviewer 9P6s (Part 2/2)**
>
> **Weakness 4.** The Competitive Normalized Fusion is a commonly adopted strategy when a network has multiple parallel branches.
>
> **Response:**  Thank you for your comment. When a network has multiple parallel branches, common strategies mostly utilize addition and concatenation [1-4]. In our work, since both parallel branches are implemented by the attention mechanism, we ingeniously leverage competition between the attention weights of both downsampling and neighboring tokens to achieve deep multi-scale feature fusion, **thereby aligning within deep dynamic interactions between local fine-grained details and global coarse-grained semantics inherent in foveal visual perception.** From the perspective of biological vision, the competitive normalized fusion can also be intuitively explained as a competition for attention between the focus and periphery. Moreover, the ablation comparison in Tab. 8 further experimentally demonstrates the effectiveness of the proposed competitive normalized fusion.
>
> [1] Z. H. Li, P. Gao, K. You, C. Yan, and M. Paul. Global attention-guided dual-domain point cloud feature learning for classification and segmentation. IEEE Transactions on Artificial Intelligence (TAI), 5(10):5167–5167, Oct. 2024b.
>
> [2] T. Zhang, X. T. Li, H. B. Yuan, S. P. Ji, and S. C. Yan. Point cloud mamba: Point cloud learning via state space model. arXiv:2403.00762, 2024.
>
> [3] Y. X. Fu, M. Lou, and Y. Z. Yu. SegMAN: Omni-scale Context Modeling with State Space Models and Local Attention for Semantic Segmentation. In Proc. IEEE/CVF Conference on Computer Vision and Pattern Recognition (CVPR), Nashville, TN, USA, Jun. 2025, pp. 19077–19087.
>
> [4] A. M. Shaker, S. T. Wasim, S. H. Khan, J. Gall, and F. S. Khan, GroupMamba: Efficient Group-Based Visual State Space Model. In Proc. IEEE/CVF Conference on Computer Vision and Pattern Recognition (CVPR), Nashville, TN, USA, Jun. 2025, pp. 14912–14922.
>
> **Weakness 5.** Most of the datasets used are pure point cloud data. To better align with the biological vision motivation, it is recommended to include more datasets with RGB information.
>
> **Response:** Thank you for your valuable comment. Additional information such as RGB can indeed better assist our methods in point cloud understanding. Therefore, as described in Appendix B.1, the input of S3DIS dataset includes RGB color and normalized position apart from point geometry coordinates, while the input of ModelNet40 and ShapeNet datasets includes normal vectors for each point apart from point geometry coordinates. After passing through the embedding layer shown in Fig. 2, these additional features are transformed into high-dimensional space to implicitly assist the network in performing higher-order semantic perception.
>
> **Weakness 1.** Although the motivation and starting point of the paper are sound, the core ideas of the proposed solution lack novelty.
>
> **Response:** Thank you for your feedback. As with the responses to Weaknesses 1-3, the fine-grained perception module does not simply update the central token by performing attention computation with k neighboring tokens, but instead collaboratively works with the downsampling tokens to capture the fine-grained details in consideration of coarse-grained features; the coarse-grained awareness module achieves the downsampling and feature aggregation of point clouds through learnable queries with cross-attention; the competitive normalized fusion provides and validates a superior approach for merging two attention branches in point cloud analysis networks. These proposed modules and their corresponding experimental results demonstrate the novelty and effectiveness of our method.
>
> Importantly, based upon the point-focused attention, our work further uses the Hilbert curve and state space model to construct the saccade process of biological vision, finally yielding a focus-then-context biomimetic model for point cloud learning.

---

> ### Author Response · Authors · 2025-11-27
> **We are open to any further discussion.**
>
> Dear Reviewer 9P6s,
>
> We sincerely appreciate the time and effort you invested in reviewing our paper. We hope that our explanations have satisfactorily addressed your concerns. As the discussion phase is ongoing, we welcome any additional comments or questions you may have regarding our response or the main paper. Please do not hesitate to contact us if further clarification is needed, and we will promptly provide the necessary information. We look forward to your valuable feedback.
>
> Best regards,
>
> Paper 21998 Authors

---

### Author Response · Authors · 2025-11-20
**A summary of the major changes made in the revised manuscript**

First of all, we would like to thank the Program Chairs, Senior Area Chairs, Area Chairs, and all the anonymous reviewers for their precious time and thoughtful review of this manuscript. The raised comments and suggestions are extremely valuable and constructive, which are very helpful for improving the quality of the manuscript. In this revision, we have considered and addressed all the comments with our utmost carefulness. In responding to the reviewers’ invaluable comments, significant changes have been made to the original manuscript.

**A summary of the major changes made in the revised manuscript is given in the following.**

- We have added attention heatmaps for PointLearner in Appendix E.1 to better understand the attention response and the advantage of the proposed method.

- We have further discussed in Appendix C.1 the advantages of using the Hilbert curve over the Z-Order curve and learnable serialization strategies in terms of spatial locality preservation, continuity, and computational efficiency.

- We have compared PointLearner with several previous state-of-the-art methods in Tab. 5 to further analyze its computational overhead, with the params, latency, and memory footprint in a single inference being used as evaluation metrics.

- We have listed detailed network architectures and training settings across different datasets in a tabular form in Appendix B.2 to better understand the model's structure and implementation.

- We have fully discussed self-supervised pretraining methods for hybrid architectures as a future research direction in Appendix F.

---

### Meta-Review · Area_Chair_mqAF · 2025-12-22

**Summary:**

After considering the rebuttal and reviewers’ post-rebuttal assessments, the consensus is that the proposed PointLearner framework is clearly motivated and demonstrates strong empirical performance across multiple point cloud benchmarks, with additional robustness and efficiency analyses. Reviewers consistently acknowledge the clarity of presentation and the effectiveness of the hybrid attention–SSM design.

The remaining disagreement centers on the degree of task-specific novelty, as some reviewers view the individual components as incremental relative to prior attention- and SSM-based methods. However, the authors’ rebuttal clarifies that the contribution lies in the system-level integration, including competitive normalization between local and downsampled tokens, point-cloud-specific induced pooling, and geometry-aware serialization for global inference. While the novelty may be incremental, no major technical or experimental issues remain unresolved.

Overall, the work meets the standard expected for a top-tier venue, with solid empirical validation and a coherent modeling framework.

**Reviewer Concerns:**

Reviewer 9P6s:
Finds the paper well motivated with strong results but remains unconvinced about the level of methodological novelty beyond existing attention and SSM hybrids.

Reviewer 5iH6:
Views the framework as reasonable and clearly presented, but questions whether the biomimetic formulation is sufficiently domain-specific to point clouds.

Reviewer ZCEa:
Acknowledges improved clarity and helpful ablations after rebuttal, while still expressing some concerns about generality and heuristic design choices.

Reviewer T5db:
Is generally positive and considers most concerns addressed, with no major issues beyond broader novelty considerations.

**Reviewer Scores:**

Reviewer 9P6s: Likely no change.

Reviewer 5iH6: Likely no change or a slight increase.

Reviewer ZCEa: Likely a modest increase.

Reviewer T5db: Likely no change.

---

### Decision · Program_Chairs · 2026-01-26

Accept (Poster)